# CoverPruneGS: Coverage-Preserving Structured Pruning for Compact 3D Gaussian Splatting from Sparse-View Monocular Videos

Yang Xiao [1]   Guoan Xu [1]   Guxue Gao [2]   Qiang Wu [1]   Wenjing Jia [1]

## Abstract

Reconstructing complete yet compact 3D Gaussian Splatting (3DGS) representations from sparse-view monocular videos remains a significant challenge. While hierarchical training with Video Frame Interpolation (VFI) improves coverage, its correlated pseudo-views and repeated merging accumulate structured, non-i.i.d. redundancy, violating the implicit independence assumptions of standard pruning methods and rendering global thresholding ineffectual. We propose **CoverPruneGS**, a coverage-preserving structured pruning framework specifically designed for hierarchical 3DGS. Our approach implements a coarse-to-fine pruning pipeline using voxel-based local diversity selection and ground-truth-guided lazy refinement via randomized dropout rendering. To ensure reliable refinement, we introduce a footprint-aware CUDA attribution mechanism. By aggregating ground-truth-aligned error degradation across Gaussian-influenced pixels, we generate faithful importance scores that enable precise, quantile-based "rescue" of essential primitives. Experimental results across multiple datasets demonstrate that CoverPruneGS substantially reduces Gaussian counts by 56.8% and significantly accelerates inference speeds, all while enhancing or maintaining the quality of novel view synthesis.

## 1. Introduction

3D Gaussian Splatting (3DGS) (Kerbl et al., 2023) has recently emerged as a powerful representation for high-fidelity

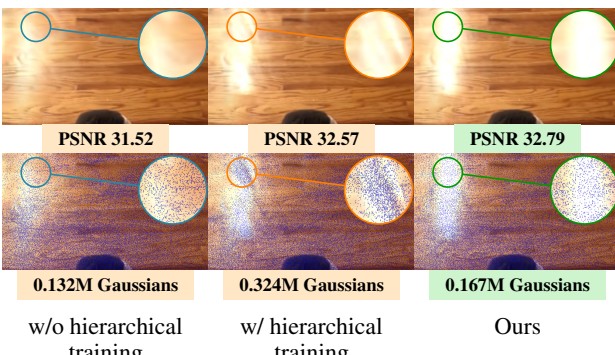

*Figure 1.* Illustration of structured redundancy induced by VFI-augmented hierarchical training. Compared to w/o hierarchical merging (left), hierarchical training (middle) improves reconstruction by expanding geometric coverage, but also induces structured, spatially clustered redundancy; our method (right) effectively suppresses this redundancy.

novel view synthesis and differentiable 3D reconstruction. However, constructing a complete yet compact 3DGS representation from sparse-view monocular long videos remains fundamentally challenging, as sparse-view supervision often induces excessive entanglement among Gaussians, causing multiple primitives to co-adapt and jointly overfit a limited set of viewpoints, leading to view-dependent appearance degradation (Chen et al., 2025).

A large body of 3DGS research has been conducted to improve sparse-view reconstruction by densifying Gaussian primitives during training, which progressively insert and refine Gaussians to alleviate geometric and appearance insufficiency caused by missing observations (Li et al., 2025; Dhiman et al., 2024; Mallick et al., 2024). Complementary sparsification-oriented approaches further alternate optimization and simplification to enforce compactness of Gaussians (Zhang et al., 2025). Meanwhile, dropout-style and probabilistic regularizations have been explored to suppress co-adaptation and redundancy under sparse supervision (Kheradmand et al., 2024; Xu et al., 2025; Chen et al., 2025; Sun et al., 2025; Liu et al., 2025; Kim et al., 2025). Although effective under available observations, these methods remain fundamentally constrained by sparse supervision and cannot resolve the missing-evidence bottleneck caused by unobserved viewpoints.

[1] Faculty of Engineering and Information Technology, University of Technology Sydney, Sydney, Australia [2] School of Computer and Software Engineering, Huaiyin Institute of Technology, Huai'an, China . Correspondence to: Wenjing Jia <Wenjing.Jia@uts.edu.au>.

*Proceedings of the 43rd International Conference on Machine Learning*, Seoul, South Korea. PMLR 306, 2026. Copyright 2026 by the author(s).

Indeed, initializing 3DGS from a single reference frame inevitably leaves large portions of the scene unobserved, leading to missing or extremely sparse Gaussian coverage in those regions. Although pre-trained priors and pseudo supervision (Liu et al., 2024; Ji & Yao, 2025; Cai et al., 2025b) can partially alleviate this issue by enriching observations, they often introduce rapidly accumulating structured, non-i.i.d. redundancy and distorted primitive distributions, as illustrated in Fig. 1. Existing efficiency-oriented pruning designs (Hanson et al., 2025a; Feng et al., 2025) mainly rely on primitive-level heuristics and global per-primitive scores. However, when many Gaussians are spatially clustered within the same local region, they often share highly similar blending weights, making such strategies redundant: aggressive pruning removes all such primitives and causes abrupt quality degradation, whereas conservative pruning preserves coverage at the cost of retaining severe redundancy (Lee et al., 2025).

Motivated by these observations, we propose **CoverPruneGS**, a coarse-to-fine structured pruning framework designed for hierarchical 3DGS with Video Frame Interpolation (VFI) augmentation, designed to suppress structured redundancy while preserving geometric coverage under sparse-view long-sequence settings. Our key design is to organize Gaussians via voxelization as a coverage-preserving prior, thereby transforming pruning in the coarse pruning stage from global per-primitive thresholding into voxel-based local representative selection, which suppresses near-duplicate Gaussians within bounded spatial neighborhoods. On top of this, we introduce a ground-truth (GT)-guided lazy refinement via randomized dropout rendering (RDR) to resolve ambiguous borderline cases: we measure GT-aligned error degradation induced by temporarily deactivating candidates and attribute the degradation back to individual Gaussians using a footprint-aware CUDA aggregation consistent with alpha compositing. This yields faithful per-Gaussian scores for quantile-based rescue, enabling robust pruning under clustered and entangled Gaussian distributions. In summary, our contributions are summarized as follows:

- We introduce **CoverPruneGS**, a coarse-to-fine structured pruning framework designed for hierarchical merging and VFI-augmented supervision.

- We propose a redundancy-aware coarse-to-fine pruning strategy that integrates voxel-based local diversity selection with a GT-guided lazy refinement based on randomized dropout rendering.

- We present a footprint-aware refinement formulation that aggregates GT-aligned error degradation over Gaussian-influenced pixels using rasterization buffers.

## 2. Related Work

### 2.1. Coverage Expansion under Sparse Views

Novel view synthesis (NVS) aims to render photorealistic images from unseen viewpoints given a limited set of observed views (Penner & Zhang, 2017). Under sparse-view monocular settings, limited observations often lead to insufficient coverage in under-observed regions, which degrades reconstruction completeness. To alleviate this missing-evidence issue, several works have been developed to enrich supervision from the input side by introducing additional views or priors (Liu et al., 2024; Cai et al., 2025a; Ji & Yao, 2025). InterPose (Cai et al., 2025a) leverages generative video models to synthesize intermediate frames with plausible camera motion, improving the robustness of pose estimation, while the HT-3DGS (Ji & Yao, 2025) integrates interpolated views with segment-wise reconstruction and progressive merging to expand geometric coverage. However, such coverage-expansion–oriented augmentation and merging strategies inevitably introduce highly correlated supervision, leading to redundant and co-adapted Gaussian primitives. To mitigate the redundancy induced by coverage expansion under sparse views, LongSplat (Lin et al., 2025) introduces an efficient octree anchor formation mechanism that converts dense point clouds into a compact anchor-based representation according to spatial density. In contrast to LongSplat, which mitigates redundancy through anchor-based reparameterization driven by spatial density, our method explicitly couples hierarchical augmentation with coverage-preserving coarse-to-fine pruning. By reasoning about redundancy within local voxel neighborhoods and refining ambiguous cases in a GT-guided manner, **CoverPruneGS** suppresses redundancy while preserving and even strengthening geometric coverage in originally sparse regions.

### 2.2. Redundancy Suppression in 3DGS

Complementary to coverage expansion, a large body of work focuses on suppressing redundancy in 3D Gaussian Splatting to obtain compact representations. Early approaches rely on hand-crafted importance criteria or global per-primitive scores to prune low-impact Gaussians, including LightGaussian (Fan et al., 2024), RadSplat (Niemeyer et al., 2025), and Taming3DGS (Mallick et al., 2024). Learnable mask-based methods further predict per-Gaussian keep/remove decisions jointly with rendering objectives, such as Compact3DGS (Lee et al., 2024) and LP-3DGS (Zhang et al., 2024). More recently, probabilistic and stochastic formulations reinterpret Gaussian simplification as a sampling or inference process, including 3DGS-MCMC (Kheradmand et al., 2024) and MH-3DGS (Kim et al., 2025). Dropout-style regularization has also been explored to mitigate co-adaptation under sparse supervision,

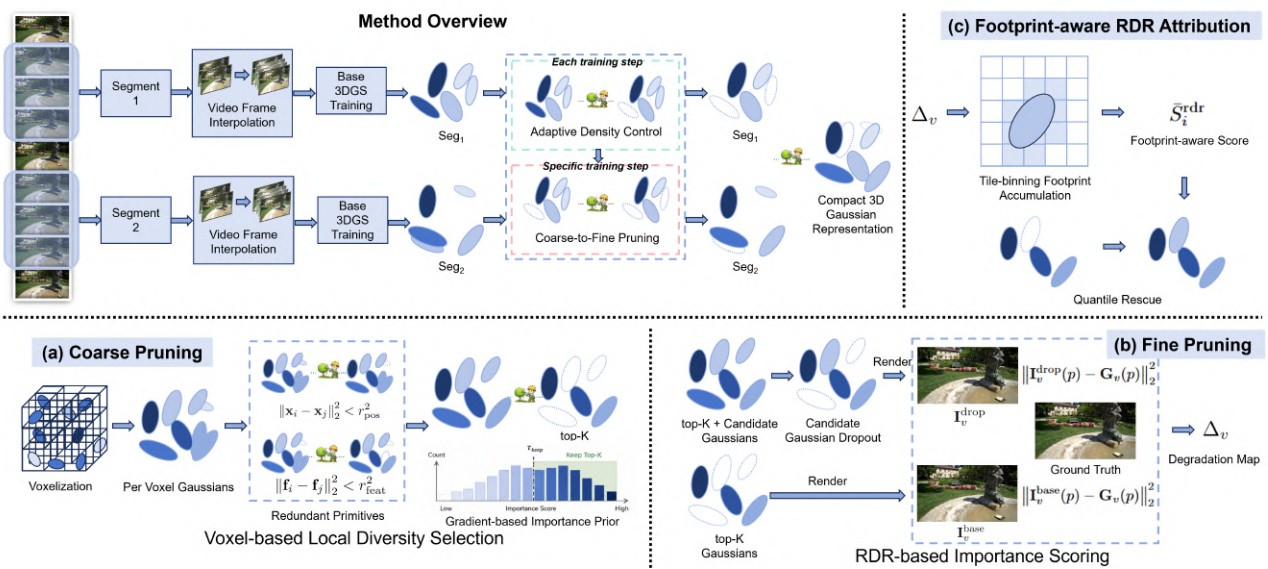

*Figure 2.* **Overview of the proposed CoverPruneGS.** Given a monocular long video under sparse-view settings, we train segment-level 3DGS models with VFI-augmented supervision and progressively merge them to expand geometric coverage. To suppress redundancy accumulated under hierarchical merging and correlated pseudo views, we design a coarse-to-fine structured pruning module into both segment densification and inter-segment fusion: Adaptive Density Control is applied at each training step, while coarse-to-fine pruning is triggered at specific steps. Coarse pruning performs voxel-based local diversity selection to remove near-duplicate Gaussians, and fine pruning applies GT-guided lazy refinement via randomized dropout rendering, where a footprint-aware CUDA attribution aggregates GT-aligned degradation $\Delta_v$ to obtain per-Gaussian RDR scores for quantile-based rescue.

as studied in DropoutGS (Xu et al., 2025) and subsequent analyses (Chen et al., 2025). Beyond single-stage pruning, a subset of recent approaches adopt a coarse-to-fine simplification strategy, where an initial conservative pruning stage removes evidently redundant Gaussians, followed by refinement steps that resolve ambiguous cases through further optimization (Fan et al., 2024; Hanson et al., 2025b).

Despite these advances, existing redundancy suppression methods largely overlook the structured redundancy induced by VFI-augmented hierarchical training. In such settings, redundant Gaussians tend to form spatially clustered, highly correlated groups, violating the common assumption that primitives are independent and identically distributed (i.i.d.). As a consequence, pruning strategies face a fundamental trade-off: aggressive pruning risks removing Gaussians that are critical for preserving geometric coverage in originally under-observed regions, whereas conservative pruning inevitably retains large amounts of redundant primitives, leading to inefficiency and limited compression.

## 3. Method

### 3.1. Overview of CoverPruneGS

The overall pipeline is illustrated in Fig. 2. Given a monocular video sequence $\{I_i\}_{i=1}^N$ with known intrinsics, our goal is to reconstruct a unified 3D Gaussian Splatting (3DGS)

representation that supports high-fidelity novel view synthesis under sparse-view and long-sequence settings. Following HT-3DGS (Ji & Yao, 2025), we first estimate a global camera pose sequence using SfM-free relative alignment, partition the video into overlapping segments, train segment-level 3DGS models, and progressively merge them into a unified scene representation. After each merge, the fused model is further optimized using VFI-generated intermediate frames to enrich supervision at previously unobserved viewpoints.

A central challenge in this setting is to suppress the rapidly accumulating redundancy introduced by hierarchical merging and correlated VFI supervision, while preserving and strengthening geometric coverage in originally underobserved regions. To address this redundancy–coverage dilemma, we design a coarse-to-fine structured pruning pipeline into both segment densification and inter-segment merging. Our key design is to organize Gaussians using voxelization and perform pruning in a structured, locality-aware manner, rather than relying on global per-primitive thresholding. By discretizing 3D space into regular voxel cells, pruning is transformed from global per-primitive thresholding into local representative selection: within each voxel, near-duplicate Gaussians are explicitly suppressed while a small set of representatives is preserved.

This voxel-wise grouping explicitly serves as a coverage-

preserving prior: by enforcing locality during pruning, it prevents under-observed regions from being eliminated by global criteria, while safely removing redundancy in over-dense areas. On top of this coarse, voxel-based selection, we introduce a fine pruning stage via GT-guided lazy refinement. Instead of committing hard decisions for all candidates, we refine only borderline cases by measuring GT-aligned error degradation under randomized dropout rendering, and attribute the degradation back to individual Gaussians using footprint-aware aggregation. This coarse-to-fine design enables robust pruning under clustered and entangled Gaussian distributions, yielding compact representations without sacrificing coverage or rendering fidelity.

## 3.2. Coarse Pruning

### 3.2.1. ADAPTIVE DENSITY CONTROL

We adopt Adaptive Density Control (ADC) (Kerbl et al., 2023) to remove evidently unqualified or low-contribution Gaussian primitives, thereby shrinking the candidate set and reducing the computational cost of subsequent redundancy-aware pruning. This step produces a base-surviving mask $\mathbf{M}_{\mathrm{base}} \in \{0,1\}^N$, where $\mathbf{M}_{\mathrm{base}}(i) = 1$ indicates that Gaussian $i$ survives the ADC filtering and is retained for further processing.

### 3.2.2. GRADIENT-BASED IMPORTANCE PRIOR

To obtain a conservative importance prior for coarse selection, we reuse the gradient-based importance measure employed in HT-3DGS (Ji & Yao, 2025), which was originally introduced for 3DGS compression (Fan et al., 2024). The importance of a given parameter $p$ is defined as:

$$S_i^{\mathrm{imp}} = \frac{1}{\sum_{i=1}^{N} H_i W_i} \sum_{i=1}^{N} \left| \frac{\partial \left( \sum_{x,y} \hat{I}_i(x,y) \right)}{\partial p} \right|, \quad (1)$$

where $\sum_{x,y} \hat{I}_i(x,y)$ denotes the sum of RGB values in the rendered image $\hat{I}_i$, and $H_i$ and $W_i$ are the height and width of $\hat{I}_i$, respectively. $p$ is a general notation representing Gaussian attributes (e.g., color, opacity, or covariance). Based on this formulation, we compute a base importance score $S_i^{\mathrm{imp}}$ for each 3D Gaussian, which serves as the retention criterion for constructing a conservative candidate set in the subsequent voxel-based local selection stage.

### 3.2.3. VOXEL-BASED LOCAL DIVERSITY SELECTION

Most existing pruning methods implicitly assume that Gaussian primitives are independently and identically distributed (i.i.d.). However, when multiple Gaussians with near-identical appearance occupy the same voxel, gradient-based importance measures tend to assign them nearly indistinguishable scores. As a result, threshold-based pruning becomes brittle: a slightly aggressive threshold removes all such Gaussians at once, causing abrupt quality degradation, whereas a conservative threshold retains all of them, leading to substantial redundancy (Lee et al., 2025). Therefore, we introduce voxel-based local diversity selection to suppress near-duplicate Gaussians accumulated in local neighborhoods after hierarchical merging and VFI-based supervision, preventing redundant primitives with highly similar geometry and appearance from co-existing in the same region.

Let $\mathbf{x}_i \in \mathbb{R}^3$ denote the 3D center of Gaussian $i$. To enforce locality explicitly, we discretize the 3D space into a regular voxel grid with cell size $\delta$, so that redundancy removal is performed within bounded spatial neighborhoods rather than globally. Specifically, each Gaussian is assigned to an integer voxel coordinate

$$\mathbf{v}_i = \left\lfloor \frac{\mathbf{x}_i - \mathbf{b}_{\min}}{\delta} \right\rfloor, \quad (2)$$

where $\mathbf{b}_{\min}$ is the minimum corner of the bounding box of the active Gaussians. Subtracting $\mathbf{b}_{\min}$ makes the indexing translation-invariant and yields non-negative, stable voxel IDs across hierarchical merging stages, while $\delta$ controls the locality granularity of our selection. After voxelization, all Gaussians sharing the same $\mathbf{v}_i$ form a local group. Within each group, we prioritize representative selection using a conservative importance criterion: Gaussians are ranked by a normalized importance score $\widetilde{S}_i^{\mathrm{imp}} \in [0,1]$ in descending order, and at most half of the Gaussians in each voxel are greedily retained.

A candidate Gaussian $i$ is considered redundant if, to avoid excessive entanglement among similar Gaussians, there exists a previously selected Gaussian $j$ in the same voxel satisfying either of the spatial proximity conditions:

$$\|\mathbf{x}_i - \mathbf{x}_j\|_2^2 < r_{\mathrm{pos}}^2 \qquad \|\mathbf{f}_i - \mathbf{f}_j\|_2^2 < r_{\mathrm{feat}}^2, \quad (3)$$

where $\mathbf{f}_i$ denotes the flattened DC spherical harmonics features of Gaussian $i$, $r_{\mathrm{pos}} = 0.015$ is the spatial proximity threshold, and $r_{\mathrm{feat}} = 0.3$ is the appearance similarity threshold.

We record two Boolean masks to characterize the coarse selection results. The coarse keep mask $\mathbf{M}_{\mathrm{coarse}}$ is defined as $\mathbf{M}_{\mathrm{coarse}} \in \{0,1\}^N$, where $\mathbf{M}_{\mathrm{coarse}}(i) = 1$ indicates that Gaussian $i$ is selected as a representative within its voxel. The redundancy mask $\mathbf{M}_{\mathrm{red}}$ is defined as $\mathbf{M}_{\mathrm{red}} \in \{0,1\}^N$, where $\mathbf{M}_{\mathrm{red}}(i) = 1$ marks Gaussian $i$ as redundant under the above similarity criteria.

## 3.3. Fine Pruning via GT-guided Lazy Refinement

Although voxel-based local diversity selection preserves the most representative Gaussian within each voxel, the remaining candidates are non-uniformly distributed, so a fixed-ratio

restoration can be unstable. Inspired by DropoutGS (Xu et al., 2025), we propose a GT-guided lazy refinement mechanism via randomized dropout-based rendering (RDR). Since dropout is performed at the primitive level, this strategy can be unfair to large-scale Gaussians that cover more pixels and contribute more substantially to the final rendering, as their impact tends to be under-estimated when attribution is conducted uniformly at the primitive level. Therefore, differing from DropoutGS's render-to-render distillation under random primitive dropout and its primitive-level attribution, our refinement anchors the dropout-induced effect to the ground-truth objective and attributes the resulting degradation over each Gaussian's effective pixel footprint using rasterization buffers and alpha-compositing weights. In particular, we estimate each candidate's contribution by measuring the increase of GT-aligned rendering error when it is temporarily deactivated in randomized dropout trials.

### 3.3.1. BORDERLINE CANDIDATES FOR REFINEMENT

Instead of refining all Gaussians, which would be computationally prohibitive when evaluating dropout-induced degradation, the lazy refinement mechanism selectively operates on borderline candidates in the coarse selection stage. These candidates pass base pruning, are non-redundant, yet are excluded by voxel-based coarse pruning, placing them in a borderline regime of representative selection and making them cost-effective targets for refinement. We first exclude redundant Gaussians from the base-surviving pool:

$$\mathbf{M}_{\mathrm{cand}} = \mathbf{M}_{\mathrm{base}} \cap \sim \mathbf{M}_{\mathrm{red}}, \tag{4}$$

and define the borderline candidate set as

$$\mathcal{B} = \{\, i \mid \mathbf{M}_{\mathrm{cand}}(i) = 1, \ \mathbf{M}_{\mathrm{coarse}}(i) = 0 \,\}, \tag{5}$$

which contains Gaussians that pass base pruning, are non-redundant, but are not selected during voxel-based coarse pruning. Here, $\sim \mathbf{M}$ denotes the element-wise complement of a binary mask. Specifically, $\sim \mathbf{M}(i)$ is set to 1 when $\mathbf{M}(i) = 0$. These borderline candidates are subsequently carried forward as the candidate pool for the RDR-based refinement (see the following section), where their contributions are evaluated via randomized dropout rendering.

### 3.3.2. RDR-BASED IMPORTANCE SCORING

For each view $v$, we render a base image $\mathbf{I}_v^{\mathrm{base}}$ under mask $\mathbf{M}_{\mathrm{coarse}}$ and compute the per-pixel squared error with respect to the ground truth image $\mathbf{G}_v$:

$$\mathbf{E}_v^{\mathrm{base}}(p) = \left\| \mathbf{I}_v^{\mathrm{base}}(p) - \mathbf{G}_v(p) \right\|_2^2. \tag{6}$$

In each randomized trial $t$, we sample a fixed-size subset $\mathcal{D}_t \subset \mathcal{B}$ with $|\mathcal{D}_t| = k$ and deactivate them by setting their opacity to zero, yielding a dropped rendering $\mathbf{I}_v^{\mathrm{drop}}$ and

$$\mathbf{E}_v^{\mathrm{drop}}(p) = \left\| \mathbf{I}_v^{\mathrm{drop}}(p) - \mathbf{G}_v(p) \right\|_2^2. \tag{7}$$

We define a GT-aligned degradation map that only penalizes error increases:

$$\Delta_v(p) = \max\left(0, \ \mathbf{E}_v^{\mathrm{drop}}(p) - \mathbf{E}_v^{\mathrm{base}}(p)\right). \tag{8}$$

The formulation of $\Delta_v$ anchors the dropout-induced change to the ground-truth reconstruction objective rather than to a render-to-render discrepancy. By comparing $\mathbf{E}_v^{\mathrm{drop}}$ against $\mathbf{E}_v^{\mathrm{base}}$ under the same GT target $\mathbf{G}_v$, the resulting signal directly reflects whether removing a candidate Gaussian harms the supervision-consistent fidelity of the view.

However, since $\Delta_{v,t}$ is defined in the image space, attributing this degradation back to individual 3D Gaussians requires a principled aggregation that respects their effective influence on the rendered pixels. Using rasterization buffers, we accumulate footprint-weighted contributions of each dropped Gaussian $i \in \mathcal{D}_t$ to $\Delta_{v,t}$ across a set of views $\mathcal{V}$. We initialize $S_i^{\mathrm{rdr}} = 0$ for all $i \in \mathcal{B}$ and update it whenever $i$ is dropped, i.e., $S_i^{\mathrm{rdr}} \leftarrow S_i^{\mathrm{rdr}} + \sum_{v \in \mathcal{V}} S_{i,v,t}^{\mathrm{fp}}$. The per-view footprint-aware contribution is defined as:

$$S_{i,v,t}^{\mathrm{fp}} = \mathrm{FootprintAgg}(i, \Delta_{v,t}), \tag{9}$$

where $\mathrm{FootprintAgg}(\cdot)$ denotes a CUDA-implemented aggregation over the pixels influenced by Gaussian $i$, normalized by accumulated coverage with a small constant $\epsilon$ (see Sec. 3.3.3). Here, $\mathcal{V}$ denotes the set of views evaluated in trial $t$.

Under fixed-$k$ sampling, each candidate Gaussian is dropped with a uniform inclusion probability $\pi = k/|\mathcal{B}|$ per trial. To obtain an unbiased estimate of the expected GT-aligned degradation, we apply Horvitz–Thompson reweighting (Horvitz & Thompson, 1952) to compensate for subsampling bias, and normalize by the number of times each candidate is dropped:

$$\bar{S}_i^{\mathrm{rdr}} = \frac{1}{c_i + \epsilon} \sum_{t:\, i \in \mathcal{D}_t} \frac{|\mathcal{B}|}{k} \sum_{v \in \mathcal{V}} S_{i,v,t}^{\mathrm{fp}}, \tag{10}$$

where $S_{i,v,t}^{\mathrm{fp}}$ denotes the footprint-aware degradation contribution of Gaussian $i$ on view $v$ in trial $t$, $c_i$ is the total number of trials in which $i$ is dropped, and $\epsilon > 0$ is a small constant for numerical stability.

### 3.3.3. FOOTPRINT-AWARE RDR ATTRIBUTION

**Footprint-aware Attribution Formulation.** To obtain a faithful 2D→3D attribution, we aggregate the GT-aligned degradation map $\Delta_{v,t}$ over each Gaussian's effective footprint in a way consistent with forward alpha compositing. Let $p = (x, y)$ denote a pixel and let $\mathcal{P}$ be the set of pixels in the rendered image. During front-to-back compositing at pixel $p$, Gaussian $i$ contributes with opacity $\alpha_i(p) \in [0, 1)$ and transmittance $T_i(p) \in (0, 1]$ (the accumulated trans-

parency before compositing $i$ at $p$). We define the blending weight $w_i(p) = \alpha_i(p)\, T_i(p)$, and compute an alpha-weighted mean score:

$$S_{i,v,t}^{\text{fp}} = \text{FootprintAgg}(i, \Delta_{v,t}) = \frac{\sum_{p \in \mathcal{P}} w_i(p)\, \Delta_{v,t}(p)}{\sum_{p \in \mathcal{P}} w_i(p) + \epsilon}. \quad (11)$$

Here, the numerator measures how much $\Delta_{v,t}$ falls on pixels effectively influenced by $i$, and the denominator normalizes by the total accumulated coverage of $i$.

**CUDA Realization via Tile-binning.** Directly iterating over all pixels for every Gaussian is prohibitively expensive. Instead, we reuse the rasterizer's tile-binning structure to traverse only *relevant* (tile, pixel, Gaussian) interactions, i.e., Gaussians whose screen-space ellipses overlap a tile. We partition the image into tiles of size $(\texttt{BLOCK\_X} \times \texttt{BLOCK\_Y})$ and denote a tile by $\tau$. For each tile $\tau$, the rasterizer provides (i) an index range $\text{ranges}[\tau] = [l_\tau, r_\tau)$ and (ii) a flattened list $\texttt{point\_list}$ such that $\{\, \texttt{point\_list}[j] \mid j \in [l_\tau, r_\tau) \,\}$ enumerates the Gaussians that may affect pixels inside tile $\tau$ (already sorted in the same front-to-back order used by forward compositing). Our CUDA kernel assigns one thread to one pixel $p$ in a tile and iterates the corresponding Gaussians in this order. For each Gaussian $i$, we evaluate its conic (ellipse) footprint at $p$ to obtain $\alpha_i(p)$, compute the blending weight $w_i(p) = \alpha_i(p)\, T$, and accumulate $w_i(p)\Delta_{v,t}(p)$ and $w_i(p)$ into the per-drop $\texttt{num}/\texttt{den}$ buffers when $i \in \mathcal{D}_t$. We update the transmittance as $T \leftarrow T(1 - \alpha_i(p))$ and early-stop once $T$ falls below a small threshold, since subsequent contributions become negligible; this matches the renderer and keeps the attribution consistent with alpha compositing.

To compute the two sums in Eq. 11 efficiently only for the dropped subset $\mathcal{D}_t$, we build a scatter lookup table $\mathbf{d}$ over the active Gaussians used by the rasterizer in trial $t$:

$$\mathbf{d}(i) = \begin{cases} \ell, & \text{if } i \text{ is the } \ell\text{-th element in } \mathcal{D}_t, \\ -1, & \text{otherwise.} \end{cases} \quad (12)$$

where $\ell \in \{1, \dots, k\}$. When processing pixel $p$, if $\mathbf{d}(i) \geq 0$ we accumulate

$$\texttt{num}[\ell] \mathrel{+}= w_i(p)\, \Delta_{v,t}(p), \qquad \texttt{den}[\ell] \mathrel{+}= w_i(p), \quad (13)$$

which correspond exactly to the numerator and denominator in Eq. 11. After all tiles are processed, we finalize $S_{i,v,t}^{\text{fp}} = \texttt{num}[\mathbf{d}(i)]/(\texttt{den}[\mathbf{d}(i)] + \epsilon)$.

**Coverage-first Scheduling and Quantile Rescue.** To ensure that all candidates are evaluated at least once under fixed-size dropout, we adopt a coverage-first sampling schedule that traverses the candidate list in contiguous blocks of size $k$ with wrap-around. After sufficient trials, we select the final rescued set by thresholding $\bar{S}_i^{\text{rdr}}$ using a

quantile $q$. Unlike fixed thresholds, which can fail across scenes with varying redundancy levels, quantile-based rescue adapts to the empirical distribution of RDR scores, ensuring stable retention behavior under different redundancy regimes. Formally, the rescued set is defined as:

$$\mathcal{R}_q = \left\{ i \in \mathcal{B} \mid \bar{S}_i^{\text{rdr}} \geq \text{Quantile}_q\left(\{\bar{S}_j^{\text{rdr}}\}_{j \in \mathcal{B}}\right) \right\}. \quad (14)$$

We denote by $\mathbf{M}_{\mathcal{R}_q} \in \{0,1\}^N$ the binary mask induced by $\mathcal{R}_q$, i.e., $\mathbf{M}_{\mathcal{R}_q}(i) = 1$ iff $i \in \mathcal{R}_q$. The final keep mask is then obtained by uniting the coarse representatives and the rescued borderline candidates. We optionally apply early stopping when the top-$K$ ranking stabilizes to reduce redundant trials:

$$\mathbf{M}_{\text{final}} = \mathbf{M}_{\text{coarse}} \cup \mathbf{M}_{\mathcal{R}_q}. \quad (15)$$

### 3.4. Loss Functions

We follow the setting of HT-3DGS (Ji & Yao, 2025) and optimize the 3DGS model using a photometric reconstruction loss between the rendered image and the supervision image, which can be either the original training image or a pseudo image. Specifically, we use a weighted combination of an $\ell_1$ term and a D-SSIM term (Kerbl et al., 2023):

$$\mathcal{L}_{\text{photo}} = (1 - \lambda)\, \mathcal{L}_1 + \lambda\, \mathcal{L}_{\text{D-SSIM}}, \quad (16)$$

where $\mathcal{L}_1$ denotes the pixel-wise $\ell_1$ loss and $\mathcal{L}_{\text{D-SSIM}} = 1 - \text{SSIM}$ encourages structural consistency between the rendered and supervision images.

## 4. Experiments

### 4.1. Implementation Details

Similar to state-of-the-art methods, we evaluate **CoverPruneGS** on standard novel view synthesis (NVS) benchmarks, following the same dataset splits and evaluation protocols adopted in previous works. All experiments are conducted on a single NVIDIA L40 GPU.

**Novel View Synthesis without SfM Preprocessing.** Following recent state-of-the-art novel view synthesis methods that without SfM preprocessing, we conduct experiments on Tanks & Temples (Knapitsch et al., 2017) and CO3D-V2 (Reizenstein et al., 2021), where we evaluate eight scenes from Tanks & Temples and five scenes from CO3D-V2 (see Appendix A.3). All compared methods are recent state-of-the-art SfM-free approaches, including CF-3DGS (Fu et al., 2024), LongSplat (Lin et al., 2025) and HT-3DGS (Ji & Yao, 2025).

**Comparisons with Gaussian Simplification Methods.** To assess the effectiveness of our Gaussian simplification strategy, we further compare **CoverPruneGS** with representative baselines and a broad range of state-of-the-art Gaussian simplification methods on three real-world

*Table 1.* Results on Tanks & Temples (Knapitsch et al., 2017). The best and second best results for each metric are color coded. All inference times are reported in minutes and seconds (mm:ss).

| Method | Venue | Metric | Church | Barn | Museum | Family | Horse | Ballroom | Francis | Ignatius | Mean |
|---|---|---|---|---|---|---|---|---|---|---|---|
| CF-3DGS | CVPR'24 | PSNR↑ | 30.23 | 31.23 | 29.91 | 31.27 | 33.94 | 32.47 | 32.72 | 28.43 | 31.28 |
| | | SSIM↑ | 0.93 | 0.90 | 0.91 | 0.94 | 0.96 | 0.96 | 0.91 | 0.90 | 0.93 |
| | | LPIPS↓ | 0.11 | 0.10 | 0.11 | 0.07 | 0.05 | 0.07 | 0.14 | 0.09 | 0.09 |
| LongSplat | ICCV'25 | PSNR↑ | 30.96 | 32.58 | 33.78 | 33.67 | 33.42 | 32.80 | 33.60 | 31.61 | 32.80 |
| | | SSIM↑ | 0.93 | 0.92 | 0.95 | 0.96 | 0.96 | 0.95 | 0.91 | 0.94 | 0.94 |
| | | LPIPS↓ | 0.10 | 0.09 | 0.06 | 0.06 | 0.06 | 0.06 | 0.15 | 0.07 | 0.08 |
| HT-3DGS | CVPR'25 | PSNR↑ | 31.43 | 34.70 | 31.58 | 34.52 | 35.82 | 34.00 | 34.01 | 30.51 | 33.32 |
| | | SSIM↑ | 0.94 | 0.96 | 0.95 | 0.97 | 0.98 | 0.97 | 0.93 | 0.94 | 0.96 |
| | | LPIPS↓ | 0.08 | 0.05 | 0.07 | 0.05 | 0.03 | 0.04 | 0.13 | 0.07 | 0.07 |
| | | #GS (M) | 1.387 | 2.164 | 1.489 | 1.760 | 1.270 | 1.757 | 1.260 | 2.268 | 1.670 |
| | | Time | 10:29 | 07:35 | 03:36 | 35:15 | 04:04 | 06:03 | 03:55 | 05:50 | 09:36 |
| Ours | | PSNR↑ | 31.55 | 34.63 | 31.82 | 34.39 | 35.85 | 34.01 | 34.05 | 30.93 | 33.40 |
| | | SSIM↑ | 0.94 | 0.96 | 0.95 | 0.97 | 0.98 | 0.97 | 0.93 | 0.94 | 0.96 |
| | | LPIPS↓ | 0.08 | 0.05 | 0.07 | 0.06 | 0.04 | 0.04 | 0.13 | 0.07 | 0.07 |
| | | #GS (M) | 0.640 | 0.703 | 0.682 | 0.647 | 0.710 | 1.073 | 0.498 | 0.821 | 0.722 |
| | | Time | 08:12 | 04:19 | 02:25 | 21:23 | 02:49 | 03:43 | 02:49 | 03:44 | 06:11 |

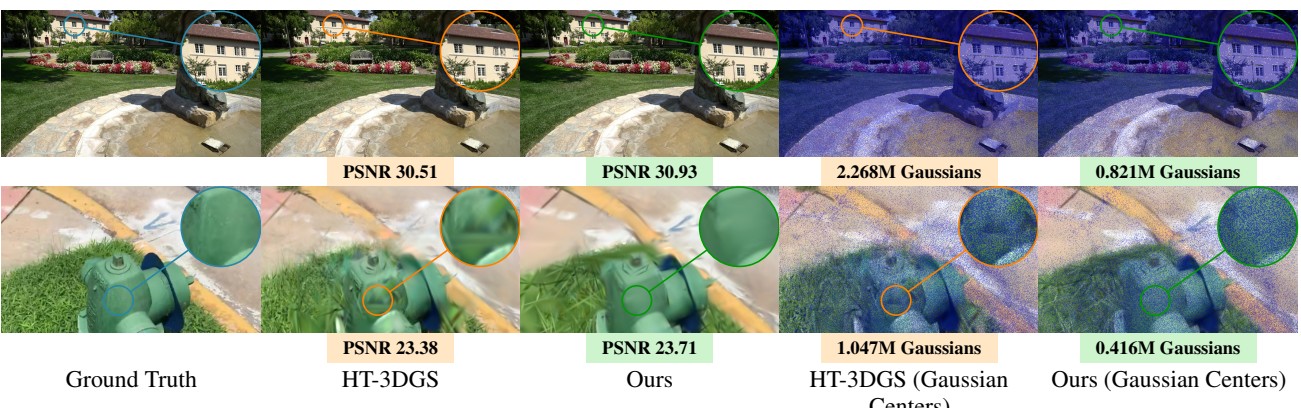

| Ground Truth | HT-3DGS | Ours | HT-3DGS (Gaussian Centers) | Ours (Gaussian Centers) |
|---|---|---|---|---|

*Figure 3.* Qualitative comparison. Rows from top to bottom correspond to two scenes: Ignatius from Tanks & Temples and Hydrant from CO3D-V2. PSNR values and Gaussian counts shown in the figure are scene-level statistics.

datasets: Mip-NeRF 360 (Barron et al., 2022), Tanks & Temples (Knapitsch et al., 2017), and Deep Blending (Hedman et al., 2018). Mip-NeRF 360 contains nine scenes (five outdoor and four indoor). For Tanks & Temples, we use the two outdoor scenes *train* and *truck*, and for Deep Blending, we use the two indoor scenes *drjohnson* and *playroom*. As primary baselines, we use vanilla 3D Gaussian Splatting (3DGS) (Kerbl et al., 2023) and Mini-Splatting (Fang & Wang, 2024). In addition, we compare against existing Gaussian simplification approaches, including Compact3DGS (Lee et al., 2024), LightGaussian (Fan et al., 2024), RadSplat (Niemeyer et al., 2025), Taming3DGS (Mallick et al., 2024), 3DGS-MCMC (Kheradmand et al., 2024), MH-3DGS (Kim et al., 2025), MaskGaussian (Liu et al., 2025), GaussianSpa (Zhang et al., 2025), and PUP 3D-GS (Hanson et al., 2025b).

### 4.2. Main Results

#### 4.2.1. COMPARISONS WITH SfM-FREE METHODS

**Quantitative Results on Tanks & Temples.** As shown in Table 1, our method consistently achieves the best or second-best performance across all scenes. Compared with prior SfM-free baselines, **CoverPruneGS** significantly improves rendering quality while producing much more compact Gaussian representations. Notably, our method reduces the number of Gaussians from an average of 1.67M in the backbone to only 0.72M, corresponding to using merely 43% of the original Gaussians. Despite this aggressive reduction, we maintain comparable image quality across all scenes. Benefiting from the reduced Gaussian count, **CoverPruneGS** achieves a clear improvement in inference efficiency, with an average inference time reduction from 9:36 to 6:11, yielding a speedup of approximately 1.55×.

*Table 2.* Unified quantitative comparison with state-of-the-art Gaussian simplification methods on novel view synthesis benchmarks. The best , second , and third best results for each metric are color coded.

| Method | Mip-NeRF360 | | | | Tanks & Temples | | | | Deep Blending | | | |
|---|---|---|---|---|---|---|---|---|---|---|---|---|
| | PSNR↑ | SSIM↑ | LPIPS↓ | #GS(M)↓ | PSNR↑ | SSIM↑ | LPIPS↓ | #GS(M)↓ | PSNR↑ | SSIM↑ | LPIPS↓ | #GS(M)↓ |
| Compact3DGS | 27.32 | 0.805 | 0.233 | 1.533 | 23.61 | 0.846 | 0.180 | 0.960 | 29.58 | 0.903 | 0.248 | 1.310 |
| LightGaussian | 27.10 | 0.800 | 0.246 | 1.090 | 23.04 | 0.822 | 0.222 | 0.625 | 27.29 | 0.877 | 0.294 | 0.752 |
| RadSplat | 27.45 | 0.811 | 0.223 | 2.184 | 23.61 | 0.847 | 0.178 | 1.053 | 29.55 | 0.903 | 0.244 | 1.515 |
| Taming3DGS | 27.31 | 0.801 | 0.252 | 0.630 | 23.95 | 0.837 | 0.201 | 0.290 | 29.82 | 0.904 | 0.260 | 0.270 |
| 3DGS-MCMC | 27.69 | 0.816 | 0.234 | 0.723 | 24.30 | 0.862 | 0.170 | 0.773 | 29.84 | 0.906 | 0.254 | 0.751 |
| MH-3DGS | 27.34 | 0.798 | 0.241 | 0.723 | 23.99 | 0.852 | 0.166 | 0.773 | 30.12 | 0.909 | 0.245 | 0.751 |
| MaskGaussian-$\alpha$ | 27.43 | 0.811 | 0.227 | 1.205 | 23.72 | 0.847 | 0.181 | 0.590 | 29.69 | 0.907 | 0.244 | 0.694 |
| GaussianSpa | 27.85 | 0.825 | 0.214 | 0.547 | 23.98 | 0.852 | 0.180 | 0.269 | 30.37 | 0.914 | 0.249 | 0.335 |
| PUP 3DGS | 26.67 | 0.786 | 0.272 | 0.273 | 22.72 | 0.801 | 0.244 | 0.156 | 28.85 | 0.881 | 0.302 | 0.246 |
| 3DGS | 27.44 | 0.811 | 0.223 | 3.176 | 23.63 | 0.848 | 0.177 | 1.831 | 29.53 | 0.904 | 0.244 | 2.815 |
| **+ Ours** | 27.89 | 0.827 | 0.211 | 1.877 | 23.88 | 0.854 | 0.170 | 1.091 | 30.08 | 0.907 | 0.241 | 1.540 |
| Mini-Splatting | 27.40 | 0.821 | 0.219 | 0.559 | 23.45 | 0.841 | 0.186 | 0.319 | 30.05 | 0.909 | 0.254 | 0.397 |
| **+ Ours** | 27.38 | 0.820 | 0.230 | 0.401 | 23.32 | 0.837 | 0.193 | 0.161 | 29.93 | 0.907 | 0.255 | 0.297 |

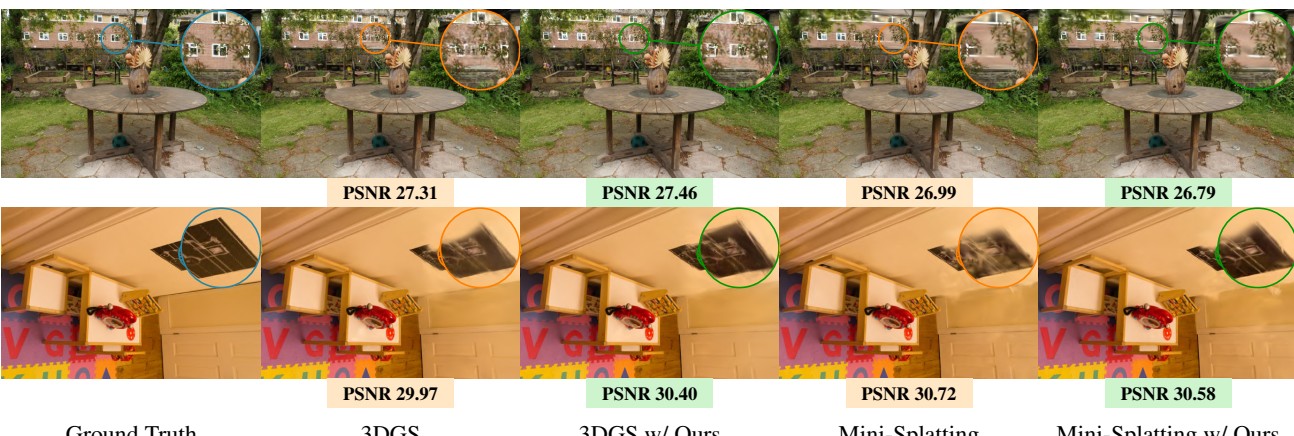

| | | | | |
|---|---|---|---|---|
| | PSNR 27.31 | PSNR 27.46 | PSNR 26.99 | PSNR 26.79 |
| | PSNR 29.97 | PSNR 30.40 | PSNR 30.72 | PSNR 30.58 |
| Ground Truth | 3DGS | 3DGS w/ Ours | Mini-Splatting | Mini-Splatting w/ Ours |

*Figure 4.* Qualitative comparison across multiple datasets. Rows from top to bottom correspond to two scenes: Garden from Mip-NeRF360 and Playroom from Deep Blending. PSNR values and Gaussian counts shown in the figure are scene-level statistics.

**Qualitative Results.** Fig. 3 shows that **CoverPruneGS** produces sharper details and a more faithful appearance while using substantially fewer Gaussians than HT-3DGS (Ji & Yao, 2025). HT-3DGS tends to exhibit local artifacts caused by over-entangled and redundant Gaussian primitives, whereas our coverage-preserving structured pruning yields cleaner edges, more stable textures, and improved visual consistency. The Gaussian-center visualizations further confirm that our method maintains a more compact and less redundant distribution, thereby avoiding local artifacts induced by excessive Gaussian entanglement. Fig. 4 further shows qualitative comparisons between 3DGS, Mini-Splatting, and their variants enhanced with our method.

### 4.2.2. COMPARISONS WITH GAUSSIAN SIMPLIFICATION METHODS

Table 2 summarizes the unified quantitative comparison on Mip-NeRF 360 (Barron et al., 2022), Tanks & Tem-ples (Knapitsch et al., 2017), and Deep Blending (Hedman et al., 2018), where **CoverPruneGS** is applied as a plug-and-play pruning module on top of different backbones. When applied to the 3DGS baseline, our method consistently achieves a favorable quality–compactness trade-off: it reduces the number of Gaussian substantially, while simultaneously improving rendering fidelity. These gains indicate that our redundancy-aware coarse-to-fine pruning effectively removes near-duplicate primitives without over-pruning informative Gaussians, leading to a more compact yet higher-quality representation. On top of the already compact Mini-Splatting baseline, our approach further reduces the number of Gaussians on all benchmarks. While the representation becomes tighter, the metrics show the expected compression–fidelity tension in this near-saturation regime: we observe mixed changes in PSNR and LPIPS, with a notable SSIM improvement on Mip-NeRF360, suggesting that our structured pruning can still suppress residual redundancy even for compact models.

*Table 3.* Ablation of the structured pruning pipeline on Tanks & Temples. We progressively add voxel-based local diversity (LD) selection and the GT-guided lazy refinement via randomized dropout rendering (RDR), and further evaluate the footprint-aware attribution used inside RDR.

| Method | Base | LD | RDR | Barn | | | | Museum | | | | Francis | | | |
|---|---|---|---|---|---|---|---|---|---|---|---|---|---|---|---|
| | | | | PSNR↑ | SSIM↑ | LPIPS↓ | #GS↓ | PSNR↑ | SSIM↑ | LPIPS↓ | #GS↓ | PSNR↑ | SSIM↑ | LPIPS↓ | #GS↓ |
| Base-Prune (Ji & Yao, 2025) | ✓ | | | 34.70 | 0.964 | 0.048 | 2.164 | 31.58 | 0.947 | 0.074 | 1.489 | 34.01 | 0.928 | 0.131 | 1.260 |
| LD (w/o rescue) | ✓ | ✓ | | 34.37 | 0.959 | 0.055 | 1.214 | 31.51 | 0.944 | 0.078 | 1.033 | 33.90 | 0.926 | 0.136 | 0.697 |
| LD (w/ global rescue) | ✓ | ✓ | | 34.60 | 0.962 | 0.051 | 1.317 | 31.57 | 0.946 | 0.076 | 1.095 | 33.95 | 0.927 | 0.135 | 0.750 |
| LD+RDR (w/o footprint) | ✓ | ✓ | ✓ | 34.58 | 0.962 | 0.052 | 0.731 | 31.79 | 0.947 | 0.075 | 0.680 | 33.93 | 0.927 | 0.137 | 0.497 |
| RDR-only | ✓ | | ✓ | 34.70 | 0.964 | 0.049 | 0.866 | 31.65 | 0.947 | 0.074 | 0.915 | 33.96 | 0.927 | 0.135 | 0.631 |
| **CoverPruneGS (Full)** | ✓ | ✓ | ✓ | 34.63 | 0.962 | 0.052 | 0.703 | 31.82 | 0.948 | 0.073 | 0.682 | 34.05 | 0.929 | 0.133 | 0.498 |

Among the most relevant coarse-to-fine baselines, Light-Gaussian (Fan et al., 2024) and PUP 3D-GS (Hanson et al., 2025c) rely on global, primitive-level importance signals that become unreliable under clustered and entangled Gaussian distributions. As a result, LightGaussian achieves only moderate compression with limited improvement in the quality–compactness trade-off, whereas PUP 3D-GS can reduce the model to an extremely low number of Gaussians but often at the cost of noticeable quality degradation. In contrast, **CoverPruneGS** couples voxel-based local representative selection with GT-guided refinement via randomized dropout rendering and footprint-aware attribution. This design enables compact representations with stable rendering quality.

### 4.3. Ablation Study

**Effectiveness of voxel-based Local Diversity (LD) Selection.** LD (see Sec. 3.2.3) is designed to suppress near-duplicate Gaussians that accumulate in local neighborhoods after hierarchical merging and VFI-augmented supervision. As shown by LD (w/o rescue), LD alone already yields a substantial reduction of Gaussian primitives compared to the base pruning baseline, indicating that local redundancy is indeed pervasive. However, the quality metrics drop consistently, suggesting that purely local representative selection can over-prune borderline yet informative Gaussians when their contributions are highly entangled.

**Global Rescue vs. GT-guided Refinement for Stabilizing LD Decisions.** To mitigate the above over-pruning risk, we compare two rescue strategies. LD (w/ global rescue) partially recovers quality by restoring a fixed global proportion of pruned candidates: specifically, we rank the pruned candidates by the gradient-based importance prior in Eq. (1) and restore the top 30% highest-scoring ones to the keep set, which improves PSNR/SSIM across all three scenes. Nevertheless, it still retains noticeably more Gaussians than the proposed refinement variants, while the metric gains remain limited. This indicates that a uniform global restoration introduces unnecessary redundancy and does not sufficiently prioritize the truly critical primitives.

**Effectiveness of RDR with Footprint-aware Attribution.** Our GT-guided lazy refinement uses randomized dropout rendering to directly measure GT-aligned degradation induced by deactivating candidate Gaussians, thereby refining ambiguous pruning boundaries with supervision-aligned signals. As shown in Table 3, introducing RDR markedly improves rendering quality while maintaining strong sparsity, validating its necessity for preventing brittle pruning under clustered and non-uniform Gaussian distributions. Further incorporating the footprint-aware attribution consistently yields a more reliable quality–compactness trade-off, highlighting that accurate pruning decisions require the attribution to respect the effective pixel footprint and alpha-compositing weights of each Gaussian. Otherwise, naive accumulation tends to mis-assign contributions among locally entangled primitives, which leads to unstable pruning behavior.

**RDR-only Refinement vs. LD+RDR Coupling.** Finally, we compare RDR-only (base + RDR) with CoverPruneGS (base + LD + RDR). While RDR-only produces stable quality with moderate sparsity, it remains less compact than the coupled design. By explicitly enforcing voxel-based diversity before refinement, CoverPruneGS further removes locally redundant primitives and achieves a consistently better compactness profile with minimal quality sacrifice.

## 5. Conclusion

We proposed **CoverPruneGS**, a structured pruning framework for hierarchical 3D Gaussian Splatting under sparse-view long-sequence settings. To address redundancy accumulated by hierarchical merging and VFI-augmented supervision, our method performs voxel-based local pruning and refines ambiguous cases with GT-guided randomized dropout rendering using footprint-aware attribution. Experiments on multiple novel view synthesis and simplification benchmarks show that CoverPruneGS yields substantially more compact 3DGS representations while maintaining or improving rendering quality and speeding up inference.

## Impact Statement

This work aims to advance efficient and high-fidelity novel view synthesis and 3D reconstruction from sparse-view monocular videos by reducing redundancy and compute/storage costs, which can benefit scalable deployment in AR/VR and embodied perception. There are many potential societal consequences of our work, none which we feel must be specifically highlighted here.

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

# A. Appendix

## A.1. Additional Implementation Details

*Table 4.* Implementation hyperparameters used in CoverPruneGS.

| Symbol | Definition | Value |
|---|---|---|
| $\delta_{\max}$ | Initial voxel size in coarse pruning | 0.05 |
| $\delta_{\min}$ | Final voxel size in coarse pruning | 0.005 |
| $\rho$ | Per-voxel keep ratio | 0.5 |
| $q$ | Quantile for RDR-based rescue | 0.3 |
| $\epsilon$ | Numerical stability constant | $10^{-6}$ |

The voxel size $\delta$ is linearly annealed from $\delta_{\max}$ to $\delta_{\min}$ over training. Coarse-to-fine structured pruning (LD prune) is triggered only at specific training steps, which occur once every 25 densification events. All remaining iterations, including intermediate densification steps without pruning, apply Adaptive Density Control (ADC) alone. Borderline candidates excluded by coarse selection are further refined using randomized dropout rendering (RDR), where the top-$q$ quantile is rescued based on footprint-aware degradation scores. All hyperparameters are fixed across datasets and determined via preliminary pilot experiments.

## A.2. Hyperparameter Analysis of Structured Pruning

### A.2.1. VOXEL SIZE SCHEDULING

*Table 5.* Hyperparameter analysis of voxel size scheduling in voxel-based local selection on Tanks & Temples.

| | Barn | | | | Museum | | | | Francis | | | |
|---|---|---|---|---|---|---|---|---|---|---|---|---|
| Voxel size | PSNR↑ | SSIM↑ | LPIPS↓ | #GS↓ | PSNR↑ | SSIM↑ | LPIPS↓ | #GS↓ | PSNR↑ | SSIM↑ | LPIPS↓ | #GS↓ |
| Fixed (0.05) | 33.04 | 0.930 | 0.111 | 0.527 | 30.66 | 0.932 | 0.096 | 0.413 | 32.92 | 0.906 | 0.177 | 0.270 |
| Fixed (0.005) | 34.65 | 0.962 | 0.051 | 0.816 | 31.57 | 0.946 | 0.076 | 0.865 | 34.00 | 0.927 | 0.135 | 0.582 |
| **Ours (0.05→0.005)** | 34.63 | 0.962 | 0.052 | 0.703 | 31.82 | 0.948 | 0.073 | 0.682 | 34.05 | 0.929 | 0.133 | 0.498 |

We further analyze the effect of voxel size scheduling in the proposed voxel-based local diversity selection. As shown in Tab. 5, using a fixed large voxel size (0.05) leads to more aggressive grouping and stronger redundancy suppression, resulting in fewer Gaussians but noticeable degradation in reconstruction quality. In contrast, a fixed small voxel size (0.005) preserves finer details and achieves stronger rendering performance, but retains substantially more Gaussians.

Our coarse-to-fine scheduling strategy (0.05→0.005) achieves a better balance between compactness and reconstruction fidelity across all scenes. The motivation is that larger voxels at early stages encourage stronger grouping of spatially clustered and near-duplicate Gaussians introduced by hierarchical merging and VFI augmentation, while smaller voxels at later stages enable finer-grained local selection and better detail preservation. This design is consistent with the progressive nature of structured pruning and improves robustness to scale variation.

In addition, voxelization in our method serves as a locality prior rather than a strict geometric partition. Since pruning decisions are jointly determined by voxel grouping, importance ranking, and the subsequent RDR-based refinement, the method remains robust to moderate voxel-size variations.

### A.2.2. PER-VOXEL KEEP RATIO

*Table 6.* Hyperparameter analysis of the per-voxel keep ratio in voxel-based local diversity selection on Tanks & Temples.

| | Barn | | | | Museum | | | | Francis | | | |
|---|---|---|---|---|---|---|---|---|---|---|---|---|
| Per-voxel keep ratio | PSNR↑ | SSIM↑ | LPIPS↓ | #GS↓ | PSNR↑ | SSIM↑ | LPIPS↓ | #GS↓ | PSNR↑ | SSIM↑ | LPIPS↓ | #GS↓ |
| 0.25 | 34.35 | 0.960 | 0.054 | 0.394 | 31.31 | 0.944 | 0.078 | 0.477 | 33.86 | 0.925 | 0.137 | 0.394 |
| 0.75 | 34.43 | 0.960 | 0.053 | 1.116 | 31.47 | 0.946 | 0.076 | 0.787 | 33.91 | 0.927 | 0.136 | 0.604 |
| **Ours (0.5)** | 34.63 | 0.962 | 0.052 | 0.703 | 31.82 | 0.948 | 0.073 | 0.682 | 34.05 | 0.929 | 0.133 | 0.498 |

Tab. 6 analyzes the influence of the per-voxel keep ratio used in local representative selection. A smaller keep ratio (0.25) produces more compact representations but may over-prune correlated Gaussians, causing slight performance degradation. Conversely, a larger keep ratio (0.75) preserves more Gaussians and improves stability, but weakens redundancy suppression and significantly increases model size.

Our default setting (0.5) achieves the best trade-off between reconstruction quality and compactness across all evaluated scenes. This result validates that retaining approximately half of the representatives within each voxel is sufficient to preserve geometric coverage while effectively suppressing structured redundancy.

### A.2.3. QUANTILE THRESHOLD FOR RDR-BASED RESCUE

*Table 7.* Hyperparameter analysis of the quantile threshold used for RDR-based rescue on Tanks & Temples.

| | Barn | | | | Museum | | | | Francis | | | |
|---|---|---|---|---|---|---|---|---|---|---|---|---|
| Quantile for RDR-based rescue | PSNR↑ | SSIM↑ | LPIPS↓ | #GS↓ | PSNR↑ | SSIM↑ | LPIPS↓ | #GS↓ | PSNR↑ | SSIM↑ | LPIPS↓ | #GS↓ |
| 0.15 | 34.57 | 0.961 | 0.053 | 0.599 | 31.41 | 0.944 | 0.079 | 0.571 | 33.92 | 0.927 | 0.137 | 0.413 |
| 0.45 | 34.53 | 0.961 | 0.054 | 0.841 | 31.46 | 0.944 | 0.076 | 0.811 | 33.87 | 0.925 | 0.138 | 0.513 |
| **Ours (0.3)** | 34.63 | 0.962 | 0.052 | 0.703 | 31.82 | 0.948 | 0.073 | 0.682 | 34.05 | 0.929 | 0.133 | 0.498 |

We further study the quantile threshold used in the RDR-based rescue stage, as shown in Tab. 7. A smaller quantile threshold (0.15) performs more conservative rescue, resulting in fewer Gaussians but slightly reduced reconstruction quality. In contrast, a larger threshold (0.45) restores more borderline candidates, increasing model size while providing limited additional quality improvement.

Our default setting (0.3) consistently achieves the best balance between rendering quality and compactness. This observation demonstrates that moderate rescue of borderline candidates is sufficient to recover informative Gaussians without reintroducing excessive redundancy.

### A.3. Quantitative Results on CO3D-V2

*Table 8.* Results on CO3D-V2 (Reizenstein et al., 2021). The best and second best results for each metric are color coded. All inference times are reported in minutes and seconds (mm:ss).

| Method | Metric | Teddybear | Hydrant | Apple | Skateboard | Bench | Mean |
|---|---|---|---|---|---|---|---|
| Nope-NeRF | PSNR↑ | 28.62 | 20.41 | 26.86 | 25.05 | 24.78 | 25.14 |
| | SSIM↑ | 0.80 | 0.46 | 0.73 | 0.80 | 0.64 | 0.69 |
| | LPIPS↓ | 0.35 | 0.58 | 0.47 | 0.49 | 0.55 | 0.49 |
| CF-3DGS | PSNR↑ | 27.75 | 22.14 | 29.69 | 27.24 | 26.21 | 26.61 |
| | SSIM↑ | 0.86 | 0.64 | 0.89 | 0.85 | 0.73 | 0.79 |
| | LPIPS↓ | 0.20 | 0.34 | 0.29 | 0.30 | 0.32 | 0.29 |
| HT-3DGS | PSNR↑ | 32.57 | 23.38 | 29.95 | 28.59 | 27.04 | 28.31 |
| | SSIM↑ | 0.93 | 0.72 | 0.87 | 0.87 | 0.77 | 0.83 |
| | LPIPS↓ | 0.14 | 0.28 | 0.19 | 0.27 | 0.30 | 0.24 |
| | #GS (M) | 0.324 | 1.047 | 0.289 | 0.291 | 0.783 | 0.547 |
| | Time | 03:57 | 05:58 | 04:57 | 04:56 | 04:50 | 04:56 |
| Ours | PSNR↑ | 32.79 | 23.71 | 30.20 | 28.62 | 27.07 | 28.48 |
| | SSIM↑ | 0.93 | 0.74 | 0.88 | 0.88 | 0.78 | 0.84 |
| | LPIPS↓ | 0.14 | 0.27 | 0.19 | 0.27 | 0.30 | 0.23 |
| | #GS (M) | 0.167 | 0.416 | 0.118 | 0.114 | 0.407 | 0.244 |
| | Time | 03:18 | 05:06 | 04:23 | 03:27 | 04:00 | 04:03 |

As shown in Table 8, **CoverPruneGS** also consistently achieves the best performance across all scenes and metrics on CO3D-V2 (Reizenstein et al., 2021). In addition to improving rendering quality, our method substantially reduces the number of Gaussians and inference time compared to HT-3DGS (Ji & Yao, 2025). Specifically, **CoverPruneGS** reduces the average number of Gaussians from 0.55M to 0.24M, using only 45% of the original primitives. Despite using less than half of the Gaussians, our method achieves consistently better rendering quality. This compact representation directly

translates to faster inference, reducing the average inference time from 4:56 to 4:03, corresponding to an inference speedup of approximately 1.22×. These results verify the effectiveness of our structured pruning in eliminating near-duplicate Gaussians accumulated through hierarchical merging and VFI-based pseudo supervision, leading to a more compact yet meaningful representation under SfM-free and long-sequence settings.

### A.4. More Qualitative Results

Figs. 5, 6, and 7 present additional qualitative novel view rendering results of different methods. Fig. 5 shows qualitative comparisons on representative scenes from Tanks & Temples (Knapitsch et al., 2017), including *Museum*, *Barn*, *Francis*, *Church*, *Family*, and *Ballroom*, illustrating differences in geometric completeness and artifact suppression under sparse-view settings. Fig. 6 further presents qualitative results on CO3D-V2 (Reizenstein et al., 2021), covering diverse object-centric scenes such as *Teddybear*, *Apple*, *Bench*, and *Skateboard*, where fine-grained structures and appearance details are consistently preserved across novel viewpoints. Finally, Fig. 7 provides complementary comparisons across Tanks & Temples, Deep Blending, and Mip-NeRF360, demonstrating the robustness of our method across diverse scenes and datasets.

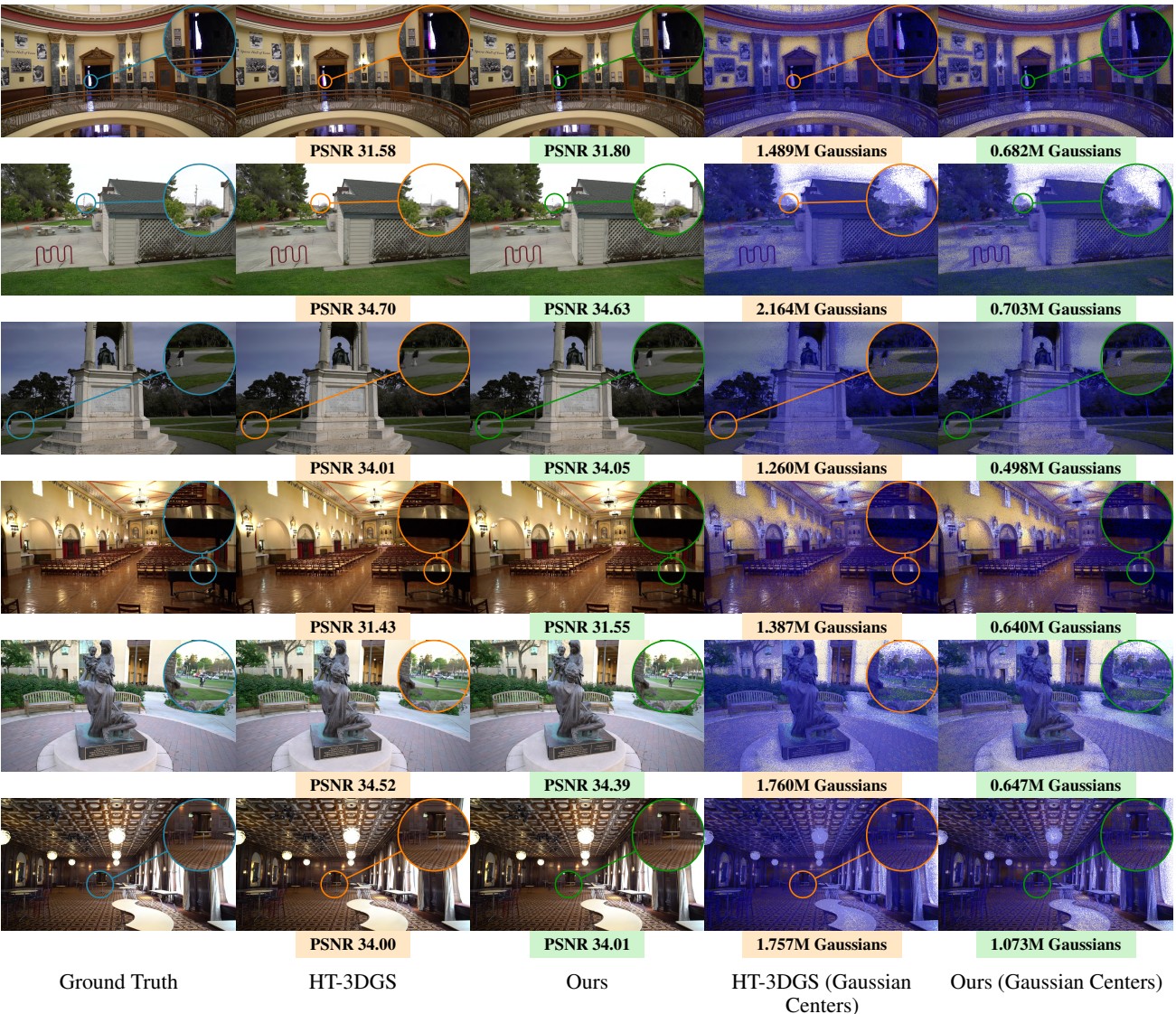

*Figure 5.* Qualitative comparison on Tanks & Temples. Rows from top to bottom correspond to six scenes: Museum, Barn, Francis, Church, Family, and Ballroom. PSNR values and Gaussian counts shown in the figure are scene-level statistics.

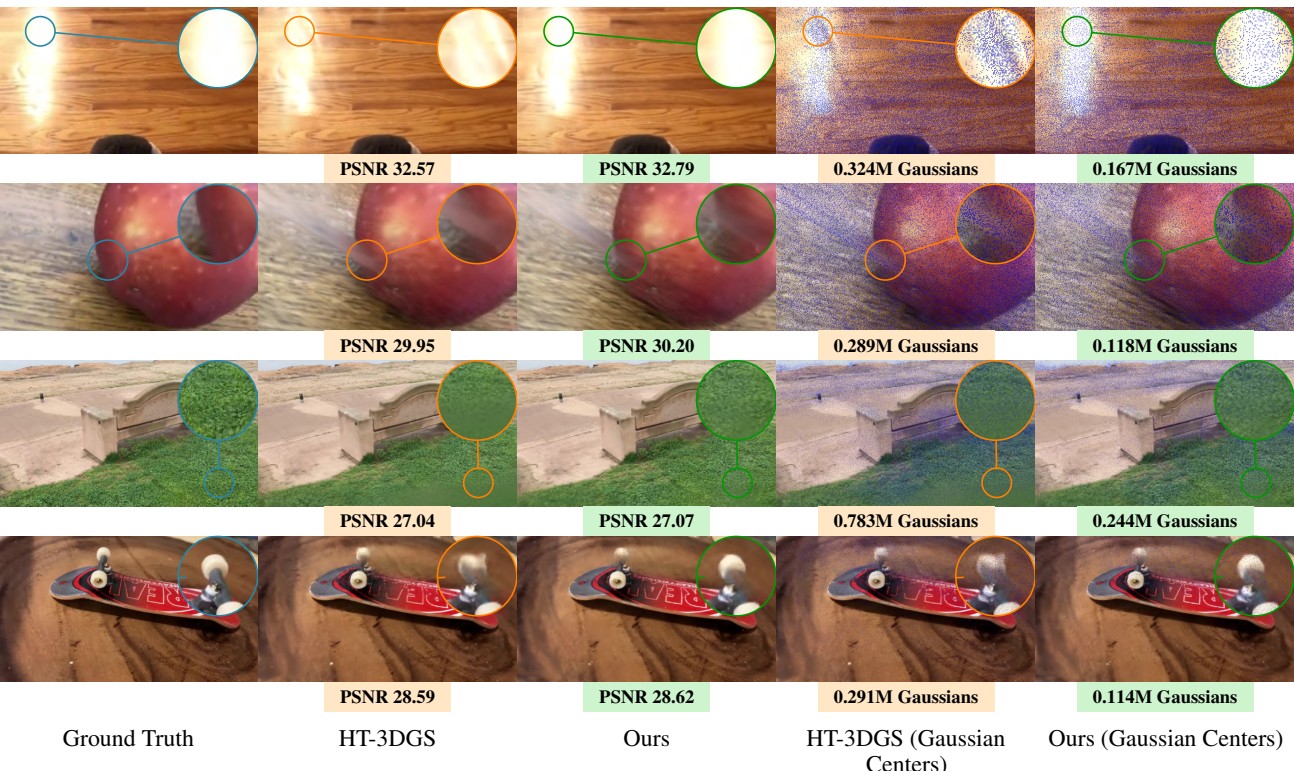

*Figure 6.* Qualitative comparison on CO3D-V2. Rows from top to bottom correspond to four scenes: Teddybear, Apple, Bench, and Skateboard. PSNR values and Gaussian counts shown in the figure are scene-level statistics.

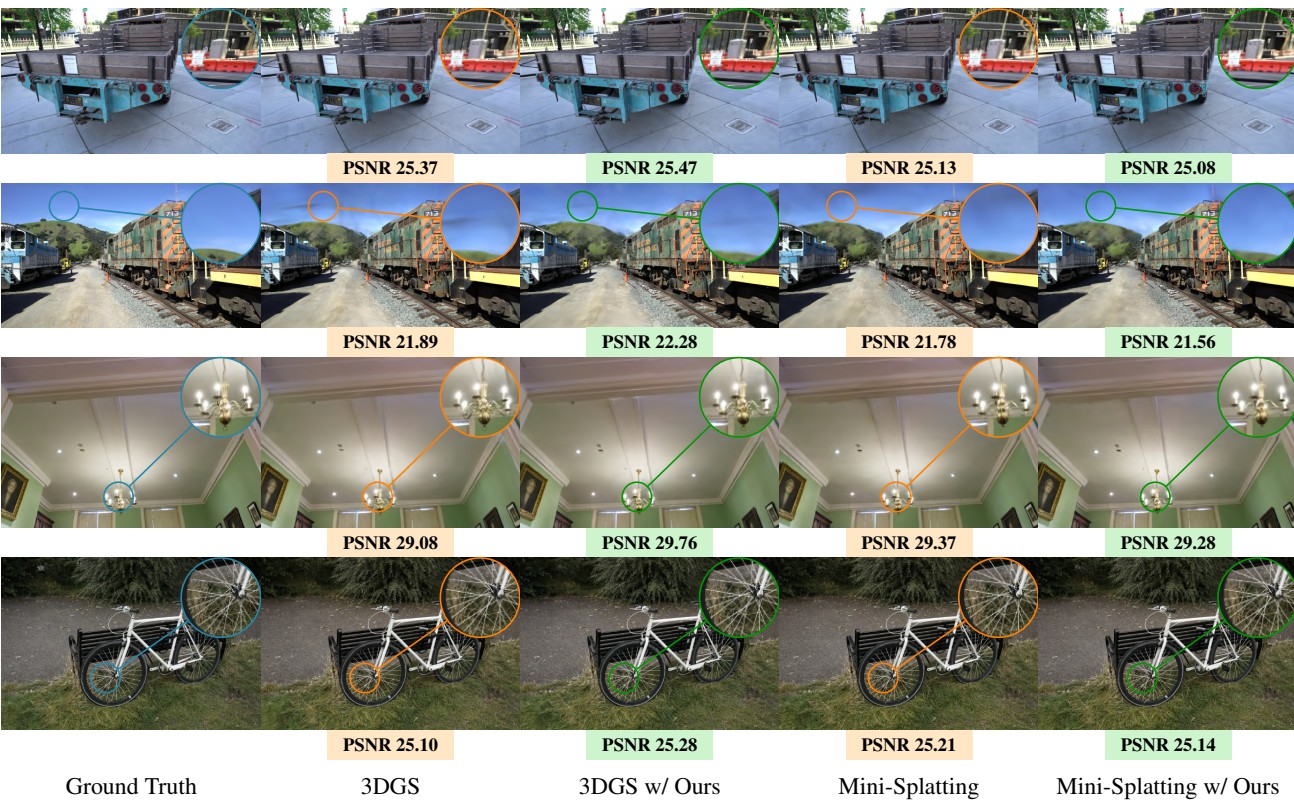

|  | PSNR 25.37 | PSNR 25.47 | PSNR 25.13 | PSNR 25.08 |
|  | PSNR 21.89 | PSNR 22.28 | PSNR 21.78 | PSNR 21.56 |
|  | PSNR 29.08 | PSNR 29.76 | PSNR 29.37 | PSNR 29.28 |
|  | PSNR 25.10 | PSNR 25.28 | PSNR 25.21 | PSNR 25.14 |
| Ground Truth | 3DGS | 3DGS w/ Ours | Mini-Splatting | Mini-Splatting w/ Ours |

*Figure 7.* Qualitative comparison across multiple datasets. Rows from top to bottom correspond to four scenes: Truck and Train from Tanks & Temples, Drjohnson from Deep Blending, Bicycle from Mip-NeRF360. PSNR values and Gaussian counts shown in the figure are scene-level statistics.

