# OpenReview forum: "CoverPruneGS: Coverage-Preserving Structured Pruning for Hierarchical 3D Gaussian Splatting from Sparse-View Monocular Videos"
_ICML.cc/2026/Conference — ICML 2026 regular_

### Official Review · Reviewer_27o7 · 2026-03-11

**Soundness:** 2
**Presentation:** 3
**Significance:** 2
**Originality:** 3
**Overall Recommendation:** 4
**Confidence:** 3

**Summary:**

This work focuses on optimizing the 3DGS representation for reconstructing complete and compact 3DGS. Specifically, this work proposes CoverPruneGS, "a coverage-preserving structured pruning framework specifically designed for hierarchical 3DGS". The coarse-to-fine structured pruning module in CoverPruneGS is served as the most important module, and includes several parts for pruning Gaussians, e.g. the coarse pruning performs voxel-based local diversity selection, the fine pruning performs randomized dropout rendering (RDR) to obtain the degradation map and per-Gaussian RDR scores. Experimental results show that CoverPruneGS achieves comparable results while using less Gaussians and time on the Tanks and Temples dataset.

**Compliance With Llm Reviewing Policy:**

Affirmed.

**Final Justification:**

I have read the rebuttal and post-rebuttal of the authors. My concerns have been adequately addressed.

I raise my score to 4. But I need to clarify that I do not have enough confidence in my assessment of the contribution of this work.

**Key Questions For Authors:**

* I'm sorry that I indeed didn't understand the part of "CUDA Realization via Tile-binning". Could you please explain it more clearly?
* I recommend the authors present a clearer explanation of ablations, since the similar performance of "CoverPruneGS (Full)" and "LD+RDR (w/o footprint)" cannot prove the effectiveness of the footprint in CUDA, as a key part of this work.

**Limitations:**

The impact statement is included, but I did not find the declaration of the limitations of this work.

**Strengths And Weaknesses:**

## Strength

* The overall writing of this work is good. The readers can easily follow up.
* The design of fine pruning is interesting for me, especially RDR and the footprint-aware score in CUDA.

## Weakness

* The experimental results do not seem to adequately demonstrate the effectiveness of the proposed method. For example, in Table 4 of A.2. Ablation Study, we can notice that the results of "LD+RDR (w/o footprint)" show limited performance degradation compared to the results of "CoverPruneGS (Full)", which means the contribution of "footprint" in the CUDA part is limited?
* I think the ablation experimental results should be included as the main experimental results in the main manuscript, rather than in the appendix.
* I think the overall approach of this work is a bit simplistic. For example, coarse pruning is essentially a statistical analysis of Gaussians. RDR is indeed an interesting idea, but the idea of ​​RDR alone may not be enough to support a complete contribution. The biggest contribution of this work may lie in the CUDA implementation, but I'm not sure if this contribution is sufficient. I will make a further judgment based on the authors' responses and the opinions of other reviewers.

---

> ### Author Rebuttal · Authors · 2026-03-30
>
> W1Q2: Thank you for the insightful comment. The seemingly small performance gap in some ablations does not indicate limited contribution, but rather reflects that footprint-aware refinement is selectively effective, particularly in challenging cases where coarse selection and primitive-level attribution fail. Instead of expensive full pixel-wise evaluation, the footprint-aware module focuses on the effective pixel footprint and alpha-compositing weights of each Gaussian, enabling more accurate and efficient per-Gaussian contribution estimation during refinement.
>
> Meanwhile, this effect is further influenced by the fact that footprint-aware scoring is applied only to a small subset of Gaussians (top quantile, e.g., 30\%), where most pruning decisions have already been determined.
>
> Furthermore, the effectiveness of this design becomes more evident when analyzing different regimes: (1) in Francis, without footprint, the LD+RDR variant suffers from performance degradation compared to Base-Prune, while footprint-aware scoring not only recovers the lost performance but further improves beyond Base-Prune, demonstrating its ability to correctly preserve truly important Gaussians; (2) in Barn, under aggressive pruning (retaining only about one third of Gaussians, e.g., 0.731M vs. 2.164M), the footprint-aware variant further reduces the Gaussian count (0.703M vs. 0.731M) while recovering part of the performance loss, indicating improved pruning precision and a better trade-off between compactness and fidelity.
>
> W2: Thank you for this valuable suggestion. We will move the ablation studies into the main manuscript in the camera-ready version to improve clarity.
>
> W3: We thank the reviewer for the thoughtful feedback. We respectfully clarify that the proposed method is not a collection of simple heuristics, but a structured design motivated by a specific failure mode in 3DGS pruning under VFI-augmented hierarchical training.
>
> First, the key challenge we address is that Gaussian redundancy in this setting is structured, spatially clustered, and non-i.i.d., rather than independent as assumed in most prior pruning methods. Under this condition, global per-primitive pruning becomes fundamentally unreliable, where entire clusters may be removed or redundant ones retained. Our coarse pruning is therefore not merely statistical filtering, but a reformulation of the pruning problem: we transform global pruning into voxel-based local representative selection, which explicitly enforces coverage preservation and resolves the instability caused by clustered redundancy, directly addressing the breakdown of i.i.d. assumptions.
>
> Second, RDR is not intended as a standalone contribution, but as a complementary refinement mechanism within a coarse-to-fine pipeline. While coarse pruning handles structured redundancy at a local level, RDR addresses ambiguous borderline cases by measuring ground-truth-aligned error degradation under stochastic dropout. Importantly, our footprint-aware attribution ensures that contribution estimation is consistent with rendering behavior (alpha compositing), which is critical for reliable refinement.
>
> Therefore, the overall framework is not a set of independent components, but a tightly coupled system, where coarse pruning resolves structured redundancy through locality-aware selection, RDR refinement resolves uncertainty via ground-truth-aligned contribution estimation, and footprint-aware attribution ensures principled and consistent scoring. Each component addresses a specific limitation of existing methods, and their combination enables robust pruning under non-i.i.d. Gaussian distributions.
>
> We will further clarify this motivation and the interplay between components in the camera-ready version to better highlight the conceptual contribution of our method.
>
> Q1: Simplified CUDA Attribution via Tile-Binning
> 1. Preprocessing
>    - Build lookup d(i): map dropped Gaussian i → index ℓ, else −1
>    - Initialize accumulators: num[ℓ]=0, den[ℓ]=0
> 2. Tile-wise parallel processing
>
>        for each tile τ:
>          for each pixel p in τ (parallel):
>            if p is outside image:
>              continue
>            Set transmittance T = 1
>            for each Gaussian i in tile range [l_τ, r_τ):
>               compute α_i(p)
>               if α_i(p) is negligible:
>                 continue
>               Compute weight w_i(p)=α_i(p)·T
>               if w_i(p) > ε and i ∈ D_t:
>                 ℓ ← d(i)
>                 num[ℓ] += w_i(p) Δ_{v,t}(p)
>                 den[ℓ] += w_i(p)
>               T ← T (1 − α_i(p))
>               if T is small:
>                 break
> 3. Normalization
>    for each ℓ:
>      S^{fp}_{i_ℓ} = num[ℓ] / (den[ℓ] + ε)
> 4. Return attribution scores for all dropped Gaussians
>
> L1: Thanks for pointing this out. We will revise the paper to include a clear discussion of limitations, such as the additional computational overhead introduced by the ground-truth-guided refinement via dropout rendering.

---

> > ### Author Rebuttal · Reviewer_27o7 · 2026-04-02
> >
> > I have read the authors' response, and I'm raising my score to 4 (actually, I'd lean towards a borderline score between 3 and 4), but I'm still unsure if the work has enough contributions and is sufficiently innovative.

---

> > > ### Author Response · Authors · 2026-04-02
> > >
> > > We sincerely thank the reviewer for carefully re-evaluating our paper and rebuttal, and we greatly appreciate the reviewer’s willingness to raise the score to 4. We also fully understand that the reviewer still has some reservation about whether the work is sufficiently innovative, and we are grateful for the opportunity to clarify this point more precisely.
> > >
> > > We would like to respectfully emphasize that the main contribution of our work is not merely a new pruning score or another simplification heuristic, but a structured pruning framework tailored to a **previously unexplored setting**: VFI-augmented hierarchical 3DGS under sparse-view supervision. In this setting, redundancy is no longer approximately independent across primitives; instead, it becomes **structured, spatially clustered, and highly correlated** due to repeated hierarchical merging and correlated pseudo-views.
> > >
> > > Recent works have begun to introduce pretrained priors, such as VFI and diffusion-based interpolation, to synthesize additional supervision (e.g., pseudo-views) and enrich observations under sparse-view inputs, thereby alleviating the missing-evidence bottleneck. However, when such pseudo supervision is combined with long-sequence hierarchical optimization, it introduces **a new challenge**: redundancy accumulates rapidly, and the distribution of Gaussian primitives becomes highly structured, spatially clustered, and strongly correlated. As a result, redundancy exhibits structured, **non-i.i.d. behavior** that fundamentally **violates the assumptions of existing pruning strategies**.
> > >
> > > Consequently, prior works on redundancy suppression mainly rely on different forms of per-Gaussian pruning or compression strategies. For example, global importance-based methods remove Gaussians using hand-crafted or learned per-primitive scores derived from contribution, sensitivity, opacity, or view statistics (e.g., LightGaussian, RadSplat, Taming3DGS).
> > >
> > > In contrast, our method is motivated by the observation that, under hierarchical VFI augmentation, many nearby Gaussians are not simply low-importance outliers, but form locally redundant clusters with similar blending behaviors. **In such cases, standard global pruning is brittle**: aggressive thresholding may remove all correlated primitives and damage coverage, while conservative pruning preserves severe redundancy. Our method addresses this overlooked issue through three coupled designs:
> > >
> > > (i) voxel-based local representative selection, which reformulates pruning from independent global scoring into locality-aware structured selection. Different from prior voxel-based designs (e.g., methods that use voxels only as a spatial partition or auxiliary structure), our method explicitly **combines voxelization with greedy representative selection** to promote local diversity while preserving global coverage. This distinction is important because previous globally scored pruning strategies treat Gaussians largely independently and cannot distinguish redundant clusters from isolated informative primitives, which may lead to local holes or excessive local accumulation and thus visible artifacts. Moreover, this stage **significantly reduces the number of ambiguous candidates** forwarded to refinement, so the additional cost of the subsequent GT-guided stage remains small.
> > >
> > > (ii) GT-guided lazy refinement via randomized dropout rendering, which revisits ambiguous borderline cases rather than making one-shot hard decisions. Since Gaussian distributions are highly non-uniform, coarse voxel-based selection may inevitably disturb the original local structure when applied conservatively for redundancy suppression. Our lazy refinement therefore uses GT as explicit guidance to recover rendering quality as much as possible **under a controlled Gaussian budget, selectively rescuing essential borderline primitives that are important for faithful rendering**.
> > >
> > > (iii) footprint-aware attribution, which measures degradation on the pixels actually influenced by each Gaussian, enabling **more faithful rescue of essential primitives**.
> > >
> > > Therefore, we believe the novelty of our work lies in introducing **a new pruning perspective** for hierarchical sparse-view 3DGS: **moving from independent primitive filtering to coverage-preserving structured redundancy suppression**. To the best of our knowledge, this specific problem formulation and solution design **have not been explicitly addressed in prior Gaussian simplification works**. Importantly, this is also supported by our empirical results, where the proposed method achieves substantially improved compactness and inference efficiency while maintaining or even improving rendering quality across benchmarks.

---

### Official Review · Reviewer_gpPZ · 2026-03-11

**Soundness:** 2
**Presentation:** 2
**Significance:** 3
**Originality:** 2
**Overall Recommendation:** 3
**Confidence:** 4

**Summary:**

This paper presents a framework for reconstructing compact 3DGS representations from sparse monocular videos. To achieve an efficient yet high-fidelity representation, the authors introduce a coarse-to-fine structured pruning module consisting of two distinct stages:

1.	Coarse Pruning: Beyond standard Adaptive Density Control, this stage partitions space into voxels and filters redundant Gaussians based on spatial proximity and color coefficient similarity. A gradient-based importance score is then employed to rank and select critical Gaussians from the non-redundant set.
2.	Fine Pruning: To recover essential details, the authors propose a ground-truth-guided lazy refinement mechanism using randomized dropout rendering. This process utilizes a footprint-aware CUDA attribution to aggregate ground-truth-aligned degradation, generating per-Gaussian RDR scores to salvage marginal but necessary Gaussians that were initially filtered.

The effectiveness of the proposed method is validated through experiments on various benchmarks. The framework demonstrates superior performance compared to both methods that bypass SfM preprocessing and existing Gaussian simplification techniques, achieving a more compact representation without sacrificing rendering quality.

**Compliance With Llm Reviewing Policy:**

Affirmed.

**Final Justification:**

Although the rebuttal is carefully prepared, in my opinion addressing key weaknesses (lack of ablation study, hyperparameter sensitivity analysis and qualitative results) indeed requires a significant update of the paper.

**Key Questions For Authors:**

1.3DGS-MCMC focuses on more efficient Gaussian distribution rather than explicit simplification (pruning); why is it categorized as a simplification method? Additionally, as Radsplat only reported results on Mip-NeRF360, could you clarify how you obtained its results for the other datasets?

2.To justify the efficiency of your method, it would be preferable to include a comparison of training/inference time against simplification baselines. Furthermore, since Taming-3DGS and 3DGS-MCMC utilize specific budgets, could you provide a comparison under the same Gaussian budget to demonstrate your method’s relative advantages?

3.The parameter selection appears overly empirical. For instance, the importance threshold is fixed at 0.5, and a uniform voxel size is used across different scenes. Could you provide ablation studies on these, or consider a more adaptive approach, such as using the median distance between initial points to determine voxel size?

4.Regarding the conflict between the text and the figure: What is the exact Gaussian mask used to generate $I_v^{base}$? Please clarify the pruning logic to resolve the discrepancy between L268 and the Figure 2 legend.

5.Your RDR-based importance seems to be an attribute derived via unbiased estimation from GT-aligned degradation maps. How does this approach compare to Perceptual-GS [1], which assigns and optimizes a learnable perceptual attribute for each Gaussian? Could you discuss the potential advantages of your estimation-based approach over their optimization-based one?

6.It is mentioned several times in the paper that “Most existing pruning methods implicitly assume that Gaussian primitives are independent and identically distributed”. Is this really the case? Why CoverPruneGS does not need this assumption but the other approaches do?

7.The role played by voxelization is not clear. Why voxel-based local diversity selection helps to suppress near-duplicate Gaussians accumulated in local neighborhoods and preserve the most representative Gaussian?

[1] Zhou, H., & Ni, Z. (2025). Perceptual-GS: Scene-adaptive perceptual densification for Gaussian splatting. arXiv preprint arXiv:2506.12400.

**Limitations:**

I believe the following aspects represent the potential limitations of this paper:

1.	Heuristic Voxel Size Selection: The reliance on an empirically predefined voxel size is a significant limitation. The framework would be more robust and adaptable if the optimal voxel size could be learned directly from the scene geometry or determined via an adaptive mechanism, rather than being fixed based on manual heuristics.

2.	Scalability to Unbounded Scenes: The current voxel-based partitioning strategy poses potential challenges for unbounded scenes (e.g., outdoor environments in Mip-NeRF360). It remains unclear how the method maintains efficiency and precision when partitioning space in regions with varying depth scales or toward the horizon, where a uniform voxel grid may become computationally or representationally suboptimal.

3.	The description of the pipeline is not clear and complete:
- As an important step in Figure 2, The merge of Seg1 and Seg2 is unexpectedly omitted in the main text.
- For clarity, Sec. 3.2.2 can be cut and moved into Sec. 3.2.3, following the introduction of voxelization.
- The ambiguity of the subscript i in the LHS and RHS of Eq. (1). Such ambiguities also occur for some other notations (v, p) and equations.

**Strengths And Weaknesses:**

Strengths
1. Originality and Significance:

•	Insightful Problem Analysis: The paper provides a compelling analysis of the limitations in current online free-viewpoint synthesis. Specifically, it correctly identifies that relying on pre-trained priors and pseudo-supervision often leads to excessive Gaussian redundancy.

•	Effective Structural Preservation: The authors offer a keen observation that spatially adjacent Gaussians often share similar contribution weights. Traditional weight-based pruning typically leads to an "all-or-nothing" dilemma—either retaining redundant clusters or destroying the underlying geometry by pruning them entirely. The proposed voxel-based representative selection effectively bypasses this issue, ensuring structural integrity while achieving a compact representation. This approach represents a logical and technically sound advancement over existing Adaptive Density Control mechanisms.

2. Soundness and Empirical Validation:

•	Extensive Comparisons: The effectiveness of the framework is validated through rigorous experiments on multiple real-world datasets. The comparisons are well-structured, covering both Gaussian simplification methods and SfM-free preprocessing methods.

•	Competitive Performance: The results demonstrate that the proposed method achieves state-of-the-art performance among SfM-free approaches. Furthermore, it remains highly competitive against specialized Gaussian simplification techniques in various scenes, successfully balancing compactness with rendering fidelity.

3. Presentation:

•	Clarity of Framework: The system pipeline is well-designed and accurately reflects the proposed framework, making the complex coarse-to-fine pruning process easy to follow.

•	Technical Depth: The inclusion of detailed mathematical formulations provides the necessary depth to understand the implementation of the structured pruning and CUDA-based attribution aggregates, aiding in the conceptual reproducibility of the work.

Weaknesses

1. Soundness

•	Incomplete Main Manuscript: The ablation studies are entirely relegated to the appendix and are not discussed within the main text. According to conference guidelines, the main manuscript must be a self-contained unit. The absence of an ablation analysis in the primary body severely weakens the empirical support for the individual components' effectiveness.

•	Lack of Hyperparameter Sensitivity Analysis: The proposed pruning and voxel-partitioning strategies involve several critical hyperparameters. However, the paper (including the appendix) lacks a sensitivity analysis for these parameters, making it difficult to assess the stability and robustness of the framework across diverse scenarios.

2. Originality & Significance

•	The qualitative results in the main text cover too few scenes. The authors should ensure that at least one representative scene from each dataset (Mip-NeRF360, Tanks & Temples, Deep Blending, and CO3D-V2) is included in the qualitative figures. Results for scenes not mentioned in the main text can be supplemented in the appendix.

3. Presentation

•	Writing Redundancy and Efficiency: Several sections are overly verbose; for instance, the detailed explanation for $b_{min}$ (L185) and the elaboration on $\tilde{M}$ (L257) could be significantly condensed. Furthermore, the textual description of "CUDA Realization via Tile-binning" (L307) would be much clearer if presented as pseudocode.

•	Technical Inconsistencies: There is a conflict between the text and the pipeline illustration (Figure 2). Line 268 states that the mask for $E_v^{base}$ is $M_{cand}$, whereas the figure legend indicates $M_{cand} \cap M_{coarse}$. This inconsistency creates confusion regarding the actual pruning logic.

•	Clerical Errors: Minor errors detract from the paper's rigor, such as Line 382 incorrectly referring to "quantitative results" when it clearly intended to say "qualitative results."

---

> ### Author Rebuttal · Authors · 2026-03-30
>
> W1W3W4W6L4L5 & W5Q4: Thank you for the helpful suggestion and pointing out the inconsistency. The correct mask used to compute $E_v^{base}$ is $M_{coarse}$, and we will incorporate this revision in the revised manuscript.
>
> W2Q3L1L2: Please refer to our response to Reviewer 2U5u W1. We agree that more adaptive strategies for voxel size selection are an interesting direction. Specifically, our voxelization is performed over the active Gaussian bounding box rather than a fixed global grid, allowing it to adapt to scene extent and remain effective even in unbounded settings. The coarse-to-fine schedule further suppresses large-scale redundancy while preserving fine details, improving robustness to scale variation. This design avoids introducing additional preprocessing or assumptions and aligns with our goal of building a general pruning framework.
>
> Q1: We include 3DGS-MCMC as a simplification-related method since it explicitly controls the Gaussian budget via a scene-dependent cap. Therefore, its optimization can be interpreted as a form of Gaussian budget-constrained simplification. Although implemented through stochastic transitions within a probabilistic framework, it effectively achieves sparsification under a fixed budget, making it relevant for comparison. For RadSplat, we adopt results from later works (e.g., MaskGaussian) for additional datasets and will clarify this in the revision.
>
> Q2: Please refer to anonymous.4open.science/r/Anonymous-F240. Our method achieves the highest rendering efficiency across all datasets, with FPS significantly outperforming prior methods, while maintaining a reasonable increase in training time due to additional optimization components. We note that methods like 3DGS-MCMC explicitly control the Gaussian budget via a scene-dependent cap, which is tuned per scene to achieve optimal performance. Instead, our method automatically determines the number of Gaussians through pruning dynamics under fixed hyperparameters, enabling a general design without per-scene adjustment. Therefore, enforcing a strict budget-aligned comparison is not directly applicable in our setting.
>
> Q5: Our RDR-based importance differs from Perceptual-GS in both formulation and objective. Perceptual-GS learns a per-Gaussian perceptual attribute jointly with optimization, which depends on the stability of learned signals. In contrast, our RDR-based importance is estimated via GT-aligned degradation under randomized dropout, providing an unbiased measure of each Gaussian’s contribution. This leads to three advantages: (1) objective alignment, as importance directly reflects reconstruction error increase; (2) reduced optimization bias, since it is not jointly learned with other attributes; (3) better handling of redundancy, as marginal contribution estimation can distinguish near-duplicate Gaussians more reliably.
>
> Q6: Most pruning methods rely on global per-primitive scoring with independent thresholding, which implicitly treats Gaussians as independent units. In Tab.5, methods without explicit dependency modeling exhibit clear redundancy accumulation under VFI-augmented training: the number of Gaussians grows to 3.9M (vs. 0.72M in ours), with frequent OOM failures and significantly slower inference (26:38 vs. 06:11). While our method avoids this implicit independence assumption: (1) in the coarse stage, voxelization groups Gaussians into local neighborhoods and reformulates pruning as representative selection, explicitly modeling local dependence; (2) in the fine stage, RDR-based refinement estimates marginal contribution under randomized dropout, naturally accounting for dependency among correlated primitives.
>
> Q7: In our method, voxelization is not merely a spatial discretization, but a structural prior that enables representative selection under local redundancy (Eq. 3). It groups Gaussians into local neighborhoods where redundancy occurs, especially under VFI and hierarchical merging. As illustrated in Fig.1, voxelization effectively suppresses the structured, spatially clustered redundancy induced by VFI. Within each voxel, pruning is reformulated as sequential representative selection rather than independent thresholding. A Gaussian is considered redundant if a previously selected neighbor is sufficiently similar in position and features, leading to a greedy selection process that suppresses near-duplicates while preserving at least one representative per region. This differs fundamentally from global pruning, which treats Gaussians independently and cannot distinguish redundant clusters from isolated informative primitives. In contrast, voxel-based selection enforces locality-aware, diversity-constrained decisions, ensuring mutual exclusivity among similar Gaussians while maintaining coverage.
>
> L3: The merge of Seg1 and Seg2 is described as “merge them into a unified scene representation”. As it is not a core contribution, we omit further details. We will clarify this step in the revision.

---

> > ### Author Rebuttal · Reviewer_gpPZ · 2026-04-01
> >
> > Although the rebuttal is carefully prepared, in my opinion addressing key weaknesses (lack of ablation study, hyperparameter sensitivity analysis and qualitative results) indeed requires a significant update of the paper.

---

> > > ### Author Response · Authors · 2026-04-01
> > >
> > > We sincerely thank the reviewer for appreciating our rebuttal, particularly our responses addressing the technical aspects of the paper. Regarding the remaining key weaknesses on presentation, we respectfully clarify that the required changes are well within the scope of a minor revision and do not involve substantial modifications to the theoretical aspects, nor do they require significant changes to the overall structure or presentation of the paper.
> > >
> > > Importantly, according to the **ICML 2026 Author Instructions** (Paper Submissions), **the final version** of each accepted paper is **allowed an extra page**. The adjustments discussed below (e.g., relocating ablations to the main text, optionally moving representative qualitative results to the main text, and supplementing hyperparameter sensitivity analysis results in the appendix) can therefore be naturally accommodated within this additional space, without requiring removal of existing core content or restructuring of the methodology, and thus do **NOT** require a significant update of the paper. In response to the reviewer’s final three concerns regarding addressing key weaknesses, the specific revision plan is as follows:
> > >
> > > (1) **Ablation studies**.
> > > We clarify that all ablation studies with detailed analysis have **already been fully provided in the appendix**. We agree that including them in the main text would improve clarity, and this can be readily achieved through simple content reorganization, with the additional page in the final version being sufficient to accommodate the ablation studies. Specifically, we will move Appendix A.2. Ablation Study (Lines 569–615) into the extra page of the final version’s main text. Since the appendix is formatted in a single-column layout, relocating it into the double-column main text will further reduce the number of occupied lines.
> > >
> > > In our actual test, after moving the Ablation Study into the main text, the content (i.e., the conclusion section) only extends from Line 427 in the right column of Page 8 to Line 461 in the right column of Page 9, **leaving 33 lines of remaining space**. Moreover, this change is purely structural and does not affect the logical completeness or experimental sufficiency of the paper.
> > >
> > > Meanwhile, according to the ICML instructions, “the Impact Statement … does not count toward the paper page limit.”
> > >
> > > (2) **Qualitative results**. We thank the reviewer for the suggestion that “the authors should ensure that at least one representative scene from each dataset (Mip-NeRF360, Tanks & Temples, Deep Blending, and CO3D-V2) ... while additional scenes can be supplemented in the appendix.” Our method is specifically designed for VFI-augmented hierarchical training. The main text, including Fig. 1 and Fig. 3, already provides representative visual results that directly validate the effectiveness of our method under this primary setting.
> > >
> > > Still, we will further include one representative scene from Deep Blending and one from Mip-NeRF360 (selected from Fig. 6 in the appendix) into the main text. Since representative scenes from Tanks & Temples and CO3D-V2 are already included in Fig. 3, we do not include additional representative scenes from these datasets in the main text.
> > >
> > > In our actual test, we include one representative scene from Deep Blending and one from Mip-NeRF360 (from Fig. 6) into the main text (rendered in double-column format), and incorporate the corresponding description into the “Quantitative Results” paragraph in Sec. 4.2.2. After this modification, the conclusion section extends from Line 461 in the right column of Page 9 (which corresponds to the layout after relocating the ablation studies into the main text, where 33 lines of space remained before filling Page 9) to Line 492, leaving 2 lines remaining before reaching the page limit (Line 494).
> > >
> > > (3) **Hyperparameter sensitivity analysis**. In the original manuscript, we stated that “All hyperparameters ... determined via preliminary pilot experiments.” Nevertheless, we agree that including such analysis can further improve the clarity of the paper. The relevant analysis has already been provided in the supplementary material of our rebuttal and fully resolves the concerns raised by Reviewer 2U5u and Reviewer gpPZ regarding hyperparameter sensitivity. These experimental results and analyses will be incorporated into the appendix in the final version, without introducing any new content beyond what has already been reviewed.
> > >
> > > Therefore, even after incorporating all the above modifications into the main text, **the final version remains within the nine-page limit**, and thus does **NOT** require any significant update to the paper.
> > >
> > > In summary, all concerns can be addressed through minor content relocation, without introducing any new content beyond what has already been reviewed, and therefore do **NOT** require a significant update of the paper; consequently, the overall quality of the paper remains well preserved.

---

### Official Review · Reviewer_ot2P · 2026-03-12

**Soundness:** 3
**Presentation:** 4
**Significance:** 3
**Originality:** 3
**Overall Recommendation:** 5
**Confidence:** 2

**Summary:**

This paper proposes a hierarchical Gaussian pruning method. In the coarse stage, voxel-partitioned Gaussians are pruned based on gradient-based importance scores. In the fine stage, a footprint-aware strategy is used to estimate the importance of each Gaussian by measuring the increase in rendering error when Gaussians are randomly dropped, enabling further pruning of less important Gaussians.The proposed method compresses the number of Gaussians while maintaining rendering performance, thereby improving efficiency. Experiments on multiple datasets demonstrate the effectiveness of the approach.

**Compliance With Llm Reviewing Policy:**

Affirmed.

**Key Questions For Authors:**

Overall, although Table 2 shows that the proposed method lags behind existing state-of-the-art approaches in terms of performance and compression efficiency, I believe the technique is sufficiently novel and provides valuable insights. This motivates my current rating. The main concerns are as described in the weaknesses above.

**Limitations:**

yes

**Strengths And Weaknesses:**

**Paper Strengths**:

1.	The paper proposes a voxel-based coarse pruning strategy to avoid the instability of threshold-based pruning.
2.	A footprint-aware RDR  is proposed to estimate the contribution of each Gaussian by accumulating the increase in pixel error when Gaussians are dropped during rendering, which is both novel and effective.
3.	The paper is well written and includes solid experimental evaluations.

**Weaknesses**:

1.	In Fig. 3 (second row), the visualization of Ours appears degraded on the grass region compared to HT-3DGS. This visually seems to be caused by over-pruning, which may suggest a potential over-pruning issue, although the overall PSNR remains higher.
2.	The results in Table 2 highlight some limitations of the proposed method compared to existing state-of-the-art approaches. For example, on the TNT dataset, Ours + 3DGS retains around 1M points, yet its rendering quality is lower than GaussianSpa, which uses only 0.26M points.

---

> ### Author Rebuttal · Authors · 2026-03-30
>
> W1: We thank the reviewer for this careful observation.
>
> We agree that, in Fig. 3 (second row), the grass region appears visually less favorable compared to HT-3DGS. At the same time, we would like to point out that our method achieves clearer reconstruction in other regions of the same scene, such as the hydrant, road markings (white paint), and the “V”-shaped ground pattern, indicating that the observed difference is region-specific rather than systematic.
>
> This behavior can be explained by the design of our RDR-based refinement. Specifically, RDR optimizes for global rendering quality under a constrained Gaussian budget, rather than enforcing uniform fidelity across all regions. As a result, the pruning process naturally prioritizes regions that contribute more significantly to overall PSNR.
>
> In this context, certain high-frequency and repetitive regions (e.g., grass) may be relatively underrepresented, which can lead to slight over-pruning and local visual artifacts. However, the corresponding error increase in these regions remains limited in terms of PSNR. Meanwhile, more structurally informative regions (e.g., object boundaries and geometric patterns) are better preserved, as they have a larger impact on global reconstruction quality.
>
> We therefore consider this example as a representative failure case, reflecting a known limitation in handling highly repetitive textures. We will clarify this point in the camera-ready version to better discuss the behavior and limitations of our method.
>
>
> W2: We thank the reviewer for this insightful observation.
>
> We agree that Table 2 reveals a gap between our method and highly optimized simplification approaches such as GaussianSpa, which achieves slightly better rendering quality with significantly fewer Gaussians on Tanks \& Temples.
>
> However, we would like to clarify that GaussianSpa and our method are designed for fundamentally different problem settings.
>
> Specifically, GaussianSpa formulates simplification as a global constrained optimization problem, where sparsity is progressively enforced and information is redistributed among remaining Gaussians during training. This design is highly effective under the assumption that Gaussian redundancy is relatively i.i.d. and globally optimizable.
>
> In contrast, our method is designed for hierarchical 3DGS with VFI augmentation, where redundancy is structured, spatially clustered, and non-i.i.d.. In such settings, we empirically observe that global pruning strategies (including importance-based or optimization-based ones) tend to produce unstable pruning decisions, as they cannot explicitly preserve coverage and local representativeness.
>
> Our approach instead introduces a coverage-preserving coarse-to-fine pruning strategy, combining voxel-based local representative selection and stochastic refinement. This design explicitly targets structured redundancy, rather than global sparsity alone.
>
> Therefore, we view the comparison in Table 2 as reflecting a trade-off between global optimal sparsification and structured robustness, where GaussianSpa excels in compression efficiency under standard settings, while our method focuses on robust pruning under structured redundancy (e.g., VFI and hierarchical merging).

---

> > ### Author Rebuttal · Reviewer_ot2P · 2026-04-02
> >
> > The authors have adequately addressed my main concerns in the rebuttal.

---

> > > ### Author Response · Authors · 2026-04-02
> > >
> > > We sincerely thank the reviewer for the careful evaluation and for acknowledging that our concerns have been adequately addressed. We truly appreciate the reviewer’s time and consideration.

---

### Official Review · Reviewer_2U5u · 2026-03-13

**Soundness:** 3
**Presentation:** 3
**Significance:** 2
**Originality:** 3
**Overall Recommendation:** 4
**Confidence:** 2

**Summary:**

This paper proposes CoverPruneGS, a pruning framework designed for hierarchical 3D GS pipelines. The authors argue that hierarchical merging creates highly correlated Gaussian clusters, which makes global pruning strategies unreliable. To address this, the method uses voxel-based local diversity selection followed by GT-guided refinement via randomized dropout rendering. Experiments show that the proposed approach reduces the number of Gaussians while maintaining comparable quality.

**Compliance With Llm Reviewing Policy:**

Affirmed.

**Final Justification:**

This authors' rebuttal addresses my concerns. After reviewing the rebuttal and the other reviewers’ comments, I will keep my score.

I want to clarify that, in my view, this work is somewhat straightforward, and I am not very familiar with some of the related developments in this specific field.

**Key Questions For Authors:**

1. Could the authors further explain the choice of hyperparameters and analyze how sensitive the method is to them? If I understand correctly, after the normalization in Equation 2, the representation becomes translation-invariant but not scale-invariant. Would the choice of voxel size therefore influence the results (e.g, indoor or outdoor scenes under different scales)?

2. The current method relies on voxel-based grouping. Would more adaptive strategies like segmentation-aware grouping be useful in such scenarios?

**Limitations:**

See weaknesses and questions above.

Overall, the paper presents an intuitive approach. My initial rating is 4: weak accept. I am open to revising my rating based on the authors’ response and other reviewers’ feedback with a better understanding of the paper.

**Strengths And Weaknesses:**

Strengths:
1. The paper is well written and easy to follow. It also explains clearly the motivation behind each component. The ablation study especially the results on reducing the number of Gaussians is convincing.
2. The design choices (e.g., GT-guided refinement via randomized dropout rendering) appear to work well and are supported by the ablation results.

Weaknesses:
1. Some hyperparameter choices appear heuristic and lack detailed analysis. For example, it would be helpful to understand how the "voxel size" schedule or the "per-voxel keep ratio" influence the results.
2. The method is designed for hierarchical pipelines with VFI. It is unclear how the proposed pruning strategy would behave in other 3DGS training settings.

---

> ### Author Rebuttal · Authors · 2026-03-30
>
> W1Q1: We thank the reviewer for this thoughtful question. For hyperparameters analyses: anonymous.4open.science/r/Anonymous-F240. Moderate hyperparameter choices, including a coarse-to-fine voxel size schedule (0.05 → 0.005), keep ratio 0.5, and quantile 0.30, achieve the best balance between reconstruction quality and model compactness, while overly aggressive or conservative settings either over-prune correlated Gaussians or retain excessive redundancy.
>
> Specifically, we would like to clarify that the voxel size schedule is not purely empirical, but is guided by both prior work and the structure of our pruning objective. We adopt a coarse-to-fine annealing schedule, inspired by the hierarchical design in MH-3DGS, where the voxel size $\delta$ is progressively reduced during training from a coarse scale ($\delta_{\max}=0.05$) to a fine scale ($\delta_{\min}=0.005$), to better support structured redundancy suppression. As shown in the hyperparameter analysis table, compared to a fixed voxel size, the proposed coarse-to-fine schedule achieves more stable performance and a better trade-off between reconstruction quality and compactness.
>
> The key motivation is that our method not only performs hierarchical organization but also explicitly targets structured redundancy. A larger voxel size at early stages promotes stronger grouping of spatially clustered and near-duplicate Gaussians, helping expose and remove redundancy introduced by hierarchical merging and VFI augmentation. In contrast, a smaller voxel size at later stages enables finer-grained local selection, improving detail preservation and reducing the risk of over-pruning. This design aligns with the progressive nature of pruning rather than relying on a single sensitive hyperparameter.
>
> Regarding sensitivity, our method is inherently robust to moderate variations in voxel size, since voxelization serves as a locality prior rather than a strict geometric partition. The pruning decision is not solely determined by voxel size, but is jointly governed by importance ranking and the subsequent RDR-based refinement, which helps correct borderline decisions. Therefore, voxel size mainly controls grouping granularity rather than acting as a hard threshold.
>
> For scale sensitivity, we agree that the normalization in Eq. (2) ensures translation invariance but not strict scale invariance. However, in practice, voxelization is performed on normalized scene coordinates (after bounding-box normalization), which implicitly mitigates scale differences across scenes. As a result, the same voxel-size schedule works consistently across both indoor and outdoor datasets in our experiments.
>
> We will further clarify this design motivation and its robustness in the camera-ready version.
>
>
> W2: We thank the reviewer for this important question.
>
> While our method is primarily motivated by hierarchical pipelines with VFI, we would like to clarify that its core design is not restricted to this specific setting, but instead targets a more general phenomenon: structured redundancy among spatially correlated Gaussians.
>
> First, the results in anonymous.4open.science/r/Anonymous-F240 show consistent reductions in the number of Gaussians across all benchmarks. Notably, even on the already compact Mini-Splatting baseline, our method further compresses the representation, while achieving FPS that significantly outperforms prior methods.
>
> Second, we observe that spatially adjacent Gaussians often share highly similar contributions, regardless of VFI. This leads traditional global or weight-based pruning to suffer from an “all-or-nothing” issue, potentially damaging geometry. In contrast, our voxel-based local representative selection enforces locality and selects representative Gaussians within each region, preserving structural integrity while achieving compactness. Therefore, while VFI-based pipelines amplify this redundancy, the underlying issue is more general, making our method applicable beyond VFI settings.
>
> Q2: We thank the reviewer for this insightful suggestion.
>
> We agree that more adaptive strategies, such as segmentation-aware grouping, could further improve pruning quality and are a promising future direction. However, our current design intentionally adopts voxel-based grouping as a generic and prior-free solution, without relying on external models or supervision.
>
> In particular, segmentation-aware grouping would require introducing external models (e.g., SAM2) or additional semantic priors, which would increase system complexity and reduce the general applicability of the method across different datasets and training pipelines. In contrast, our voxel-based grouping is lightweight, operates purely in 3D space, and is directly compatible with standard 3DGS pipelines. It leverages the observation that redundancy is typically spatially clustered, allowing voxel-based locality to effectively group redundant primitives without semantic priors.

---

> > ### Author Rebuttal · Reviewer_2U5u · 2026-04-03
> >
> > This rebuttal addresses my concerns. After reviewing the rebuttal and the other reviewers’ comments, I will keep my score.

---

> > > ### Author Response · Authors · 2026-04-03
> > >
> > > We sincerely thank the reviewer for the thoughtful evaluation and for acknowledging our responses. We greatly appreciate the reviewer’s time and constructive feedback.

---

### Decision · Program_Chairs · 2026-04-30

**Decision:**

Accept (regular)

**Comment:**

Initially this paper received mixed scores. After rebuttal, three of the reviewers agree with positive scores with one accept and two weak accepts, while there is one reviewer gives a weak reject. The AC carefully checked the paper as well as the rebuttal. As the reviewer didn't point out any specific unresolved concerns after rebuttal, the AC still recommends a decision of accept. The authors should carefully revise the paper accordingly.